# TokenDrop: Efficient Image Editing by Source Token Drop with Consistency Regularization

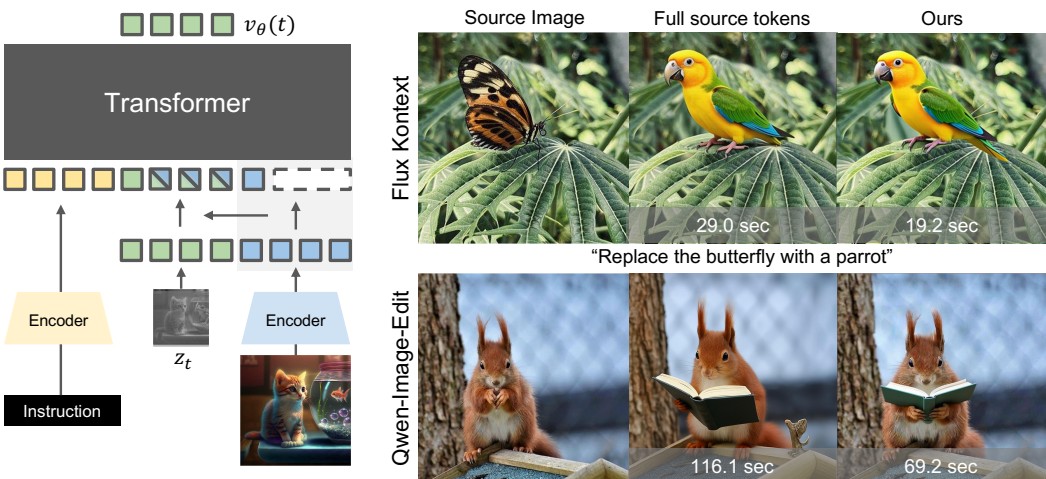

Figure 1: Overview of our method and representative results. (Left) We adaptively omit redundant source tokens to accelerate transformer-based image editing. To mitigate information loss from drop, we introduce a source consistency regularization to flow ODE that restores the influence of dropped tokens. (Right) Our approach yields more efficient and consistent editing and is applicable to various transformer-based image editing models. The number in the gray area indicates the inference time. Image size: 1024×1024.

## ABSTRACT

Text-based image editing has recently been reinterpreted in large multimodal transformers as conditional generation, where source image tokens are concatenated with text and noise tokens as conditioning inputs. While effective, this design introduces substantial computational overhead in attention layers. To mitigate this drawback, we present an efficient text-based image editing method called *TokenDrop* by dropping source tokens partially, where the selection of tokens to drop is adaptively guided by difference between the source and the clean estimate. Importantly, by reformulating the flow ODE as a latent optimization problem, we can reflect information of dropped tokens to the solution of regularized optimization. Thanks to the closed form solution, this optimization does not introduce additional computational cost. Across FluxKontext and Qwen-Image-Edit, our training-free method achieves an average 22.4% improvement in inference speed on PIEBench, while better preserving non-edited regions. The method delivers up to 1.8× speedup at $1024^2$ resolution and 2× speedup at $2048^2$ resolution.

## 1 INTRODUCTION

Inspired by the recent advances in flow and diffusion models for text-driven image generation from Gaussian noise to complex images, here we focus on text-guided image editing that modifies a source image according to a textual instruction while preserving irrelevant regions. We distinguish

our focus from "generative editing", which generates a new image preserving an object's identity in a new context (Batifol et al., 2025). Our work centers on direct image-to-image transformation.

Existing image editing approaches fall into inversion-based and inversion-free methods. Inversion-based methods reconstruct a latent noise from the source image before re-generating under the target prompt (Song et al., 2021; Su et al., 2023; Park et al., 2024; Kim et al., 2024; Rout et al., 2025), or exploit cached features during the inversion process (Tumanyan et al., 2023). These methods suffer from ODE step-size sensitivity because errors from approximating the inversion ODE accumulate when the step size is not sufficiently small. Consequently, they cannot easily reduce the number of ODE steps without sacrificing accuracy, which requires finer integration for more precise inversion (Mokady et al., 2023; Wallace et al., 2023; Meiri et al., 2023; Ju et al., 2024). In addition, they incur roughly double the computational cost, since both inversion and re-generation must be performed. Inversion-free methods avoid this doubled computation by applying a forward diffusion process to obtain an intermediate sample (Meng et al., 2021), by directly guiding the source image during editing (Hertz et al., 2023; Nam et al., 2024), or by simulating an ODE trajectory between the source and target images (Kulikov et al., 2024; Kim et al., 2025a).

More recently, large-scale multimodal transformers have unified generation and editing by processing text and source image tokens jointly (Xiao et al., 2025; Tan et al., 2024; Deng et al., 2025; Wu et al., 2025; Batifol et al., 2025). This paradigm builds on the idea introduced by InstructPix2Pix (Brooks et al., 2023), which first demonstrated conditioning a diffusion model on both the source image and editing instruction through supervised training. Although highly effective, this design introduces substantial computational overhead because the quadratic complexity of self-attention makes prepending thousands of source tokens prohibitively expensive. For instance, FluxKontext (Batifol et al., 2025) requires 29 seconds per editing of 1024×1024 image on an A100 GPU, compared to 13 seconds for standard noise to image generation.

Our work begins from the observation that not all source tokens are required for editing. In non-edit regions that are irrelevant to editing instruction and should be unmodified, source tokens primarily serve to enforce consistency with the original image, yet processing them still incurs substantial computational cost. To address this inefficiency, we reformulate the flow ODE as a latent optimization problem. In this formulation, computation on redundant tokens is replaced with a regularized ODE that enforces consistency of non-edit region, while only tokens that carry essential editing information are preserved in input token sequence. We also show that token selection should be guided by difference between the source and clean estimate, which gives rise to adaptive masking. Random token dropping under regularization collapses the process to the source image, eliminating edits, whereas residual-based selection preserves editing capability while regularizing unedited regions.

To our knowledge, our method is the first token pruning framework designed for conditional diffusion sampling in image editing. Our method is training-free and transfers the information of pruned tokens into the ODE dynamics without introducing additional neural networks for compression. Empirically, for PIEBench dataset, the proposed method increases inference throughput by average 22.4% over the vanilla model with 1024×1024 resolution (comparable in pixel count to HD), while maintaining the editing quality and providing more consistent editing with source context. Importantly, the efficiency gains grow larger at higher resolutions, demonstrating the scalability of this method. For instance, we can achieve 2× faster editing for 2048×2048 resolution image (comparable in pixel count to 4K UHD) when edited region is relatively small.

## 2 BACKGROUND

### 2.1 LINEAR FLOW MODELS

Suppose we have access to samples from the source distribution $q(\boldsymbol{x}_1)$ and the target distribution $p(\boldsymbol{x}_0)$, which consists of the independent coupling $\pi_{0,1}(\boldsymbol{x}_0, \boldsymbol{x}_1) = q(\boldsymbol{x}_1)p(\boldsymbol{x}_0)$. In this paper, we assume that $q(\boldsymbol{x}_1) := \mathcal{N}(0, \mathbf{I})$ and that $p(\boldsymbol{x}_0)$ represents an image distribution.

The flow model (Lipman et al., 2023) defines a flow $\psi_t(\boldsymbol{x}_1) := \boldsymbol{x}_t$ and the corresponding velocity at $\boldsymbol{x}_t$ as $\boldsymbol{v}(\boldsymbol{x}_t) = \dot{\psi}_t(\boldsymbol{x}_1)$, where $\dot{\psi}$ denotes the derivative of $\psi$ with respect to $t$. To explicitly express the velocity, we can define the conditional flow $\psi_t(\boldsymbol{x}_1|\boldsymbol{x}_0)$ and the corresponding velocity $\boldsymbol{v}(\boldsymbol{x}_t|\boldsymbol{x}_0)$. Among various conditional flows, the linear flow (Karras et al., 2022; Liu et al., 2023) is widely used

and is defined as $\psi_t(\boldsymbol{x}_1|\boldsymbol{x}_0) := (1-t)\boldsymbol{x}_0 + t\boldsymbol{x}_1$ with corresponding velocity $\boldsymbol{v}(\boldsymbol{x}_t|\boldsymbol{x}_0) = \boldsymbol{x}_1 - \boldsymbol{x}_0$. We can parameterize the conditional velocity using a neural network $\theta$, and the training objective, referred to as the conditional flow matching loss (Lipman et al., 2023), is given by

$$\min_\theta \mathbb{E}_{t,(\boldsymbol{x}_0,\boldsymbol{x}_1)\sim\pi_{0,1}} \|\boldsymbol{v}_\theta(\boldsymbol{x}_t,t) - (\boldsymbol{x}_1 - \boldsymbol{x}_0)\|^2 \tag{1}$$

where the gradient is equivalent to flow matching loss. Consequently, we can generate $\boldsymbol{x}_0$ from $\boldsymbol{x}_1$ by solving the ordinary differential equation (ODE) with the trained velocity model:

$$d\boldsymbol{x} = \boldsymbol{v}_\theta(\boldsymbol{x}_t,t)dt \tag{2}$$

where $dt < 0$. Importantly, this ODE can be decomposed into a clean estimate and a noise estimate using Tweedie's formula (Kim et al., 2025b):

$$d\boldsymbol{x} = (1 - t - dt)\hat{\boldsymbol{x}}_{0|t} + (t + dt)\hat{\boldsymbol{x}}_{1|t} \tag{3}$$

where

$$\hat{\boldsymbol{x}}_{0|t} := \boldsymbol{x}_t - t\boldsymbol{v}_\theta(\boldsymbol{x}_t,t) = \mathbb{E}[\boldsymbol{x}_0|\boldsymbol{x}_t] \tag{4}$$

$$\hat{\boldsymbol{x}}_{1|t} := \boldsymbol{x}_t + (1-t)\boldsymbol{v}_\theta(\boldsymbol{x}_t,t) = \mathbb{E}[\boldsymbol{x}_1|\boldsymbol{x}_t] \tag{5}$$

denote the clean estimates and the noise estimates relatively. Although the flow model is defined in latent space with an encoder $\mathcal{E}$ and a decoder $\mathcal{D}$, which consists of an auto-encoder $\boldsymbol{z} = \mathcal{E}(\boldsymbol{x}) = \mathcal{E}(\mathcal{D}(\boldsymbol{z}))$, we can use the same decomposition without loss of generality.

## 2.2 TRANSFORMER-BASED IMAGE EDITING METHODS

Recently, image generation and editing have been unified under a single or mixture of transformers (Xiao et al., 2025; Batifol et al., 2025; Wu et al., 2025; Deng et al., 2025). These models extend transformer diffusion frameworks (Peebles & Xie, 2023; Esser et al., 2024), where text embeddings and noise tokens are concatenated to predict the velocity field of the flow ODE. Unified editing methods further incorporate source image tokens as conditioning inputs, enabling strong multimodal alignment but substantially increasing sequence length. Transformer-based editing models are commonly trained with conditional flow matching (Lipman et al., 2023), where the objective is defined as

$$\min_\theta \mathbb{E}\|\boldsymbol{v}_\theta(\boldsymbol{z}_t,t,\boldsymbol{z}_{\text{src}},\boldsymbol{c}) - (\boldsymbol{z}_{\text{src}} - \boldsymbol{z}_{\text{tgt}})\|^2 \tag{6}$$

where $\boldsymbol{z}_{\text{src}} := \mathcal{E}(\boldsymbol{x}_{\text{src}})$, $\boldsymbol{z}_{\text{tgt}} := \mathcal{E}(\boldsymbol{x}_{\text{tgt}})$, and $\boldsymbol{c}$ denotes editing instruction embedding. While effective, this design requires multi-modal attention over very long sequences, making inference slow and computationally expensive. This scalability issue highlights a key limitation of existing unified approaches and motivates more efficient alternatives.

## 3 TOKENDROP: SOURCE TOKEN DROP WITH REGULARIZATION

In this paper, we consider image editing models using Multimodal diffusion transformers (MMDiT) (Esser et al., 2024) that incorporate text embeddings, noise tokens and source image tokens in the input sequence. Suppose the noise tokens at time $t$ are denoted by $\boldsymbol{z}_t \in \mathbb{R}^{L \times d}$, and the source image tokens are represented as $\boldsymbol{z}_{\text{src}} \in \mathbb{R}^{L \times d}$, where $L$ is the number of tokens and $d$ is the token embedding dimension. We will not alter text or time embedding. Let $M_t \in \mathbb{R}^L$ denotes the binary mask at time $t$, indicating the source tokens to drop. In other words, if the $i$-th elements of $M_t$ is 0, we will omit the $i$-th source token when computing velocity at time $t$. Finally, $0 \leq \lambda_t < 1$ denotes the source token drop ratio where $N := \lceil \lambda_t L \rceil$ elements of $M_t$ are zeros. When dropping source tokens, we apply the same index slicing to their positional embeddings, so the preserved tokens retain their original positional information without reindexing.

### 3.1 ANALYSIS ON RANDOM TOKEN DROP

A key observation in this paper is that the complete set of source image tokens is not always necessary for effective editing. To examine this, we analyze the effect of randomly drop tokens for Flux Kontext and Qwen-Image-Edit. Specifically, we set $\lambda_t$ as a constant for all $t$ and randomly select $N$ indices for $M_t$ to be zero for each $t$. We use the PIEBench dataset (Ju et al., 2024), which contains 700 natural and synthetic images, accompanied by editing instructions and pairs of source and target prompts.

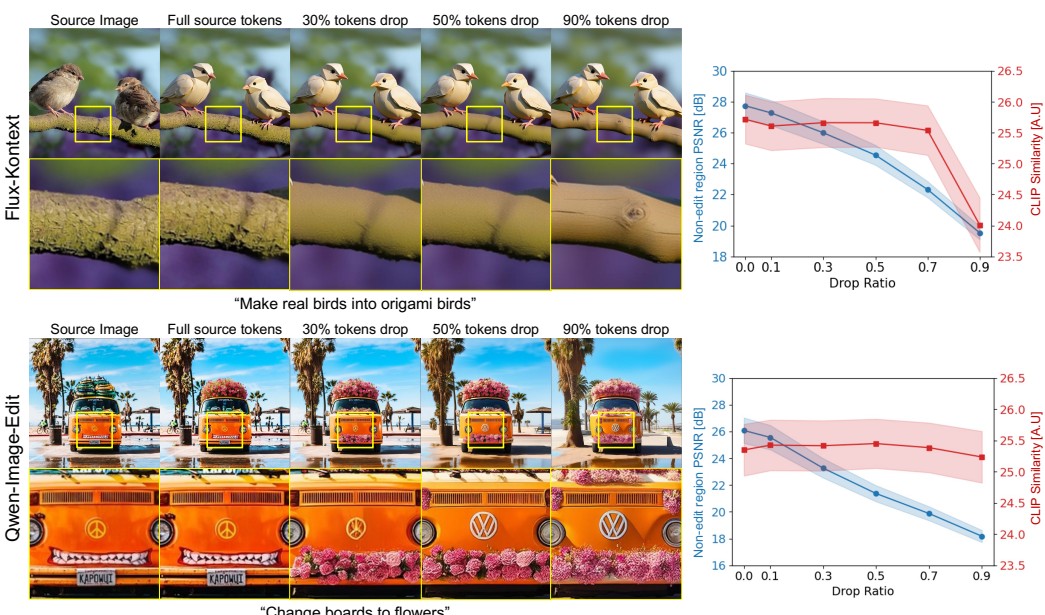

Figure 2: Effect of random token drop. (Left) Token dropping preserves text alignment but rapidly degrades consistency of non-editing area. (Right) Quantitative metrics is aligned with qualitative results. Shaded regions indicate $0.1 \times$ standard deviation across samples.

Figure 2 shows editing results under difference values of $\lambda_t$. The qualitative results indicate that the edited image remain aligned with the editing instruction even when up to 90% of the source tokens are dropped. However, higher drop ratios cause over-smoothing in unedited regions, which are not intended to be changed by instruction, leading to a loss of consistency and fine details. The quantitative results on the right panel of Figure 2 confirm this trend. CLIP similarity with the target prompt remains largely stable across drop ratios, but PSNR between unedited region and the source image decreases sharply as more tokens are omitted. Without dedicated mechanisms, the results show a clear source-efficiency trade-off under token dropping, which in turn motivates compensating for the discarded source information during flow-ODE sampling.

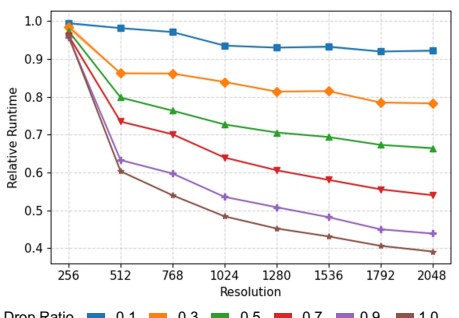

Figure 3: Relative runtime across resolutions and drop ratios. Token dropping provides greater savings at higher resolutions, where attention dominates the computation.

Another important observation is that the efficiency gains from source token dropping becomes larger at higher image resolutions. When applied to larger images, the same drop ratio yields greater runtime reductions. Figure 3 illustrates this effect. At lower resolutions, components such as feed-forward layers, and embeddings account for a larger share of the runtime, so dropping tokens provides only moderate efficiency gain. At higher resolutions, attention dominates the computation, and token dropping directly reduces this bottleneck. As a result, the measured relative runtime compared to full-token editing decreases steadily with increasing image size. This feature makes our approach increasingly advantageous for high-resolution image editing, where computational cost is the primary bottleneck.

## 3.2 MITIGATING THE SOURCE INFORMATION LOSS

Building on the findings of the previous section, we propose a training-free method to adaptively maintain source information when dropping source image tokens. Inspired by the success of Dream-Sampler (Kim et al., 2024), which reformulated the reverse sampling process as a regularized op-

timization problem, here we propose a method to explicitly enhances consistency of non-edit area using a proper regularization term. Specifically, the decomposed flow ODE in Eq. (3) can be expressed as the following optimization problem:

$$\boldsymbol{z}^* = \arg\min_{\boldsymbol{z}} \|\boldsymbol{z} - \hat{\boldsymbol{z}}_{0|t}\|^2 + \eta_t R(\boldsymbol{z}) \tag{7}$$

$$\boldsymbol{z}_{t-1} = (1 - t - dt)\boldsymbol{z}^* + (t + dt)\hat{\boldsymbol{z}}_{1|t} \tag{8}$$

where $\hat{\boldsymbol{z}}_{0|t}$ and $\hat{\boldsymbol{z}}_{1|t}$ are obtained from Eq. (4) and $\eta_t > 0$ controls the strength of regularization. Notably, when $\eta_t = 0$, the procedure reduces to the standard sampling process without any additional regularization.

Since image editing models are trained as conditional flow models that predict conditional velocity for source image, the absence of conditioning leads to inaccurate velocity estimation. This, as observed in the previous section, causes unintended deviation from the source image in regions that should be preserved. To address this issue, we introduce a source consistency regularization term:

$$R(\boldsymbol{z}) := \|\boldsymbol{z} - \boldsymbol{z}_{\mathrm{src}}\|^2, \tag{9}$$

which penalizes deviations of the sampling trajectory from source image. Importantly, because the source image latent $\boldsymbol{z}_{\mathrm{src}}$ remains accessible regardless of token drop, incorporating this regularization term does not incur additional computation overhead. This leads to the following closed-form solution of Eq. (7)

$$\boldsymbol{z}^* = \frac{1}{1 + \eta_t}\hat{\boldsymbol{z}}_{0|t} + \frac{\eta_t}{1 + \eta_t}\boldsymbol{z}_{\mathrm{src}}, \tag{10}$$

which corresponds to a linear interpolation between $\hat{\boldsymbol{z}}_{0|t}$ and $\boldsymbol{z}_{\mathrm{src}}$.

This regularization term should exert a stronger influence during the early stages of sampling and gradually diminish over time, since excessive penalization from the source image can otherwise suppress meaningful edits. Accordingly, we set the weight $\eta_t = \sigma_t$, which monotonically decreases from 1 to 0 throughout the sampling process. When $t = 1$, $\boldsymbol{z}^*$ reduces to the average of $\hat{\boldsymbol{z}}_{0|t}$ and $\boldsymbol{z}_{\mathrm{src}}$, while as $t \to 0$, $\hat{\boldsymbol{z}}_{0|t}$ dominates. This guarantees that the final output remains only marginally influenced by $\boldsymbol{z}_{\mathrm{src}}$, thereby allowing meaningful edits to be preserved. Importantly, we compute $\boldsymbol{z}^*$ only to the positions of dropped source tokens, where insufficient source information is available. If source consistency regularization is indiscriminately applied to all tokens, including those that already contain adequate source information, the editing model would suppress meaningful modifications and instead bias the output toward reproducing the original source image. Accordingly, the updated estimate is defined as

$$\tilde{\boldsymbol{z}}_{0|t} = M_t \odot \hat{\boldsymbol{z}}_{0|t} + (1 - M_t) \odot \boldsymbol{z}^* \tag{11}$$

where $\odot$ denotes a Hadamard product. Finally, the sampling step proceeds by adding deterministic noise according to Eq. (3):

$$\boldsymbol{z}_{t-1} = (1 - t - dt)\tilde{\boldsymbol{z}}_{0|t} + (t + dt)\hat{\boldsymbol{z}}_{1|t}. \tag{12}$$

### 3.3 ADAPTIVE MASK GENERATION

While we introduce a novel approach to regulate the sampling trajectory and enhance source consistency, the design of the mask is also critical for ensuring that the proposed regularization remains compatible with editing models. In particular, the following proposition demonstrates that random masking is not suitable in combination with our regularization.

**Proposition 1** (Pathwise Convergence with Random Mask). *Let $\boldsymbol{v}(\boldsymbol{z}) : \mathbb{R}^{L \times d} \to \mathbb{R}^{L \times d}$ be locally Lipschitz. With random binary mask $M_t \in \{0, 1\}^{L \times d}$, a fixed source image latent $\boldsymbol{z}_{\mathrm{src}} \in \mathbb{R}^{L \times d}$ and time $t \in (0, 1]$, the solution of expected-mask ODE in Eq. (12) $\boldsymbol{z}_t$ satisfies*

$$\lim_{t \to 0} \boldsymbol{z}_t = \boldsymbol{z}_{\mathrm{src}} \tag{13}$$

This result shows that random masking with source consistency regularization tends to collapse the editing process back toward the source image, thereby reducing editing fidelity.

To address this, we require a more principled rule for constructing the drop mask $M_t$. A desirable mask should preserve the sampling trajectory with full source tokens as closely as possible. Thus, we analyze the difference between the reference sampling trajectory in Eq. (2) and the regularized sampling trajectory in Eq. (12), leading to the following error bound.

**Proposition 2** (Error bound between two trajectories). *Let $\boldsymbol{v}(\boldsymbol{z}) : \mathbb{R}^{L \times d} \to \mathbb{R}^{L \times d}$ be locally Lipschitz. For any $t \in [t_0, t_1]$ with $0 < t_0 < t_1 \le 1$, the error between trajectory $\boldsymbol{z}_{t_0}$ (Eq. (12)) and $\boldsymbol{z}_{t_0}^{\mathrm{ref}}$ (Eq. (2)) satisfies*

$$\|\boldsymbol{e}_{t_0}\|^2 \le C(t_1, t_0)\|\boldsymbol{e}_{t_1}\|^2 + \int_{t_0}^{t_1} \frac{C(t, t_0)}{t^2}\|\boldsymbol{r}_t\|^2 dt. \tag{14}$$

*where $\boldsymbol{e}_{t_0} = \boldsymbol{z}_{t_0} - \boldsymbol{z}_{t_0}^{\mathrm{ref}}$, $C(t, t_0)$ is a time-dependent variable and $\boldsymbol{r}_t = \frac{\eta_t(1 - M_t)\odot(\hat{\boldsymbol{z}}_{0|t} - \boldsymbol{z}_{\mathrm{src}})}{1 + \eta_t}$.*

The proof is deferred to the Appendix. The upper bound consists of the initial discrepancy and the cumulative error caused by deviations of the clean estimates $\hat{\boldsymbol{z}}_{0|t}$ from the source image $\boldsymbol{z}_{\mathrm{src}}$. To make the bound smaller, the mask should suppress tokens with large deviations. In practice, instead of integrating over time, we adopt a greedy strategy based only on the current difference $\boldsymbol{r}_t$. Concretely, the mask is defined as $M_t = \mathbf{1}(D_t > \tau_{\lambda_t})$ where $\mathbf{1}$ denotes the indicator function, $D_t := |\hat{\boldsymbol{z}}_{0|t} - \boldsymbol{z}_{\mathrm{src}}|$, and the threshold $\tau_{\lambda_t}$ is chosen so that exactly $N := \lceil \lambda_t L \rceil$ source tokens with the smallest $D_t$ values are dropped.

Intuitively, this threshold separates edited region, which involves larger change from source image, from non-edit region, that should be the same as source image. Then, the adaptive mask drops non-edit region from input token sequence and replace it by source consistency regularization, which reduces the computational cost. As in the random mask case, one could use a fixed token drop ratio $\lambda_t$, but this is not always appropriate because area of non-edit region depends on images and the type of editing. For example, setting $\lambda_t = 0.9$ for a background change may fail to reduce the bound sufficiently, since tokens with large deviations could still be dropped.

To address this, we determine the drop ratio adaptively for each sample. We use triangle thresholding (Zack et al., 1977), which can effectively separate edited and non-edited regions in $D_t$ even when the histogram of difference is

---

**Algorithm 1** Regularized Sampling

**Require:** Editing model $\boldsymbol{v}_\theta$, Source Image $\boldsymbol{x}_{\mathrm{src}}$, Encoder $\mathcal{E}$ and Decoder $\mathcal{D}$, Regularization weight $\eta_t$, Instruction embedding $\boldsymbol{c}$

$\boldsymbol{z}_{\mathrm{src}} \leftarrow \mathcal{E}(\boldsymbol{x}_{\mathrm{src}}), \boldsymbol{z}_t \leftarrow \mathcal{N}(0, \mathbf{I})$
$M_t \leftarrow \mathbf{1} \in \mathbb{R}^L$
**for** $t : 1 \to 0$ **do**
    $\boldsymbol{v}(\boldsymbol{z}_t) \leftarrow \boldsymbol{v}_\theta(\boldsymbol{z}_t, \boldsymbol{c}, M_t \odot \boldsymbol{z}_{\mathrm{src}})^1$
    $\hat{\boldsymbol{z}}_{0|t} \leftarrow \boldsymbol{z}_t - t\boldsymbol{v}(\boldsymbol{z}_t)$
    $\hat{\boldsymbol{z}}_{1|t} \leftarrow \boldsymbol{z}_t + (1 - t)\boldsymbol{v}(\boldsymbol{z}_t)$
    $M_t \leftarrow \mathrm{AdaptiveMask}(\hat{\boldsymbol{z}}_{0|t}, \boldsymbol{z}_{\mathrm{src}}, t)$
    $\boldsymbol{z}^* \leftarrow M_t \odot \hat{\boldsymbol{z}}_{0|t} + (1 - M_t) \odot \boldsymbol{z}_{\mathrm{src}}$
    $\tilde{\boldsymbol{z}}_{0|t} \leftarrow \frac{1}{1 + \eta_t}\hat{\boldsymbol{z}}_{0|t} + \frac{\eta_t}{1 + \eta_t}\boldsymbol{z}^*$
    $\boldsymbol{z}_t \leftarrow (1 - t - dt)\tilde{\boldsymbol{z}}_{0|t} + (t + dt)\hat{\boldsymbol{z}}_{1|t}$
**end for**
$\boldsymbol{x}_0 \leftarrow \mathcal{D}(\boldsymbol{z}_0)$

---

not clearly multi-modal. Unlike Otsu's variance-based method (Otsu et al., 1975), triangle thresholding is particularly effective for skewed or unimodal histograms where the object peak is weak or elongated. Moreover, it is computationally inexpensive, so the additional cost of applying it during diffusion sampling is negligible compared to velocity prediction. To control the trade-off between efficiency and performance, we add a bias term and define the threshold as

$$\tau_{\lambda_t}(\omega) = \mathrm{TriangleThreshold}(D_t) - \omega\sigma_{D_t} \tag{15}$$

where $\omega \in \mathbb{R}$ controls the performance-efficiency trade-off and $\sigma_D$ denotes the standard deviation of $D_t$. This bias lowers the threshold more when the difference map is relatively flat, ensuring that fewer tokens are dropped under high uncertainty, while in sharper, peaked maps the adjustment is small and the behavior stays close to the baseline. For $1024 \times 1024$ images on an A100 GPU, masking takes 0.24s on average, compared with 0.57s for velocity prediction.[2] Finally, since $\hat{\boldsymbol{z}}_{0|t}$ is unreliable in early steps of the ODE, we use all tokens during the first few iterations to get a better initial mask estimation. The Algorithm for the proposed method is described in Algorithm 1 and 2.

## 4 EXPERIMENT

The proposed method is applicable to flow-based models that use transformers and leverage source image tokens as conditioning for image editing as depicted in Figure 1. We demonstrate its effectiveness on Flux Kontext and Qwen-Image-Edit, two recently introduced transformer-based editing

---

[1]We omit tokens of $\boldsymbol{z}_{\mathrm{src}}$ where the elements of $M_t$ are zeros.

[2]Measured during adaptive masking; both runtimes vary across ODE steps, and we report averages.

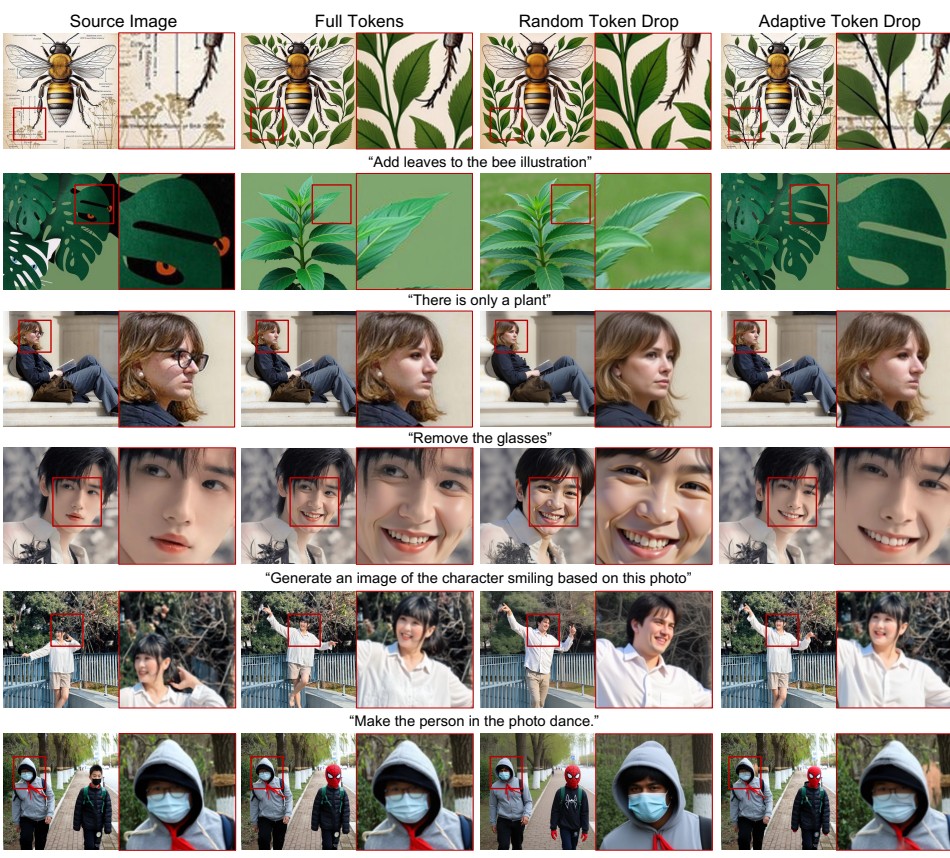

Figure 4: Qualitative comparison for Flux Kontext on PIEBench and GEdit-Bench. Editing instructions are shown below each sample. Compared with vanilla editing, the proposed method produces equivalent or more consistent results. Enlarged boxes emphasize the consistency with source image.

models with two image editing benchmarks, PIEBench (Ju et al., 2023) and GEdit-Bench (Liu et al., 2025). Additional details are provided in the Appendix B. Experiments were run on A100 GPUs.

## 4.1 EFFECTIVENESS OF ADAPTIVE MASKING

The proposed method selects tokens to drop adaptively based on difference between the clean estimate and the source image. To test its effectiveness, we compare adaptive masking with a random masking without regularization under matched drop ratios and full-token image editing. For fairness, the random strategy reuses the per-sample pruning budgets produced by adaptive masking. For Flux Kontext, we set $\omega = 0.4$ and use all source tokens for two initial ODE steps among 28 NFEs. For Qwen-Image-Edit, we set $\omega = 0.1$ for PIEBench and $\omega = 0.3$ for GEdit-Bench. We use all tokens for four initial steps among 50 NFEs. All other settings, including the classifier-free guidance scale, follow the default configurations. Input images are resized to $1024 \times 1024$ using bicubic interpolation, as required by Flux Kontext.

Table 1 presents quantitative results on PIEBench and GEdit-Bench. Definitions of each metric is described in Appendix B. With the same token drop ratios, adaptive masking consistently outperforms random masking in source consistency while maintaining text–image alignment comparable to the vanilla model. Using Qwen-Image-Edit, for instance, adaptive masking drops an average of 65% of source tokens yet reduces background PSNR by only 0.3% and CLIP similarity by 0.6% relative to full-token editing. In contrast, random masking leads to a substantial 23% drop in background PSNR. The runtime in Table 1 is averaged over all samples. Depending on the image and editing task, reductions can be larger, as shown in Figure 1, where our method achieves $1.5\times$ faster editing. At $2048 \times 2048$ resolution, we can achieves $2\times$ faster editing as shown in Section D.2.

| Model | Strategy | | PIEBench | | | | | | | | GEdit-Bench | | | |
|---|---|---|---|---|---|---|---|---|---|---|---|---|---|---|
| | | Runtime | PSNR↑ | LPIPS↓ | DINO↓ | MSE↓ | SSIM↑ | CLIP↑ | CLIP-edit↑ | Runtime | DINO↓ | Q_SC↑ | Q_PQ↑ | Overall↑ |
| | Source Image | - | ∞ | 0.000 | 0.000 | 0.000 | 1.000 | 23.19 | 20.09 | - | 0.000 | 0.276 | 7.488 | 0.244 |
| Flux Kontext | Random w/o reg | 23.72 | 23.79 | 0.120 | 0.063 | 0.011 | 0.820 | **25.89** | 22.78 | 23.72 | 0.054 | **7.033** | 7.056 | 6.630 |
| | Adaptive w/o reg | | 23.26 | 0.135 | 0.065 | 0.012 | 0.798 | 25.73 | 22.74 | | 0.056 | 6.856 | **7.175** | **6.554** |
| | Adaptive w/ reg | | 26.75 | 0.066 | **0.042** | **0.007** | **0.912** | 25.57 | 22.47 | | **0.032** | 6.100 | 7.143 | 5.842 |
| | Full Tokens | 29.01 | **27.76** | **0.063** | 0.050 | 0.009 | **0.912** | 25.86 | **22.79** | 29.01 | 0.039 | 6.291 | 7.142 | 5.909 |
| Qwen-Image-Edit | Random w/o reg | 71.12 | 19.98 | 0.160 | 0.066 | 0.019 | 0.732 | **25.50** | 22.56 | 97.02 | 0.075 | **7.917** | 7.403 | **7.586** |
| | Adaptive w/o reg | | 18.50 | 0.192 | 0.075 | 0.024 | 0.696 | 25.31 | **22.65** | | 0.087 | 7.904 | 7.434 | 7.585 |
| | Adaptive w/ reg | | 26.00 | **0.067** | **0.039** | **0.008** | **0.912** | 25.19 | 22.11 | | **0.053** | 7.415 | 7.043 | 7.004 |
| | Full Tokens | 116.1 | **26.07** | 0.088 | 0.052 | 0.013 | 0.872 | 25.35 | 22.48 | 116.1 | 0.075 | 7.876 | **7.440** | 7.536 |

Table 1: Quantitative results of Flux Kontext and Qwen-Image-Edit on PIEBench and GEdit-Bench. Average runtime (seconds) and metrics are reported. For GEdit-Bench, Qwen2.5-VL is used for evaluation. **Bold** indicates the best and underline indicates the second best except source image.

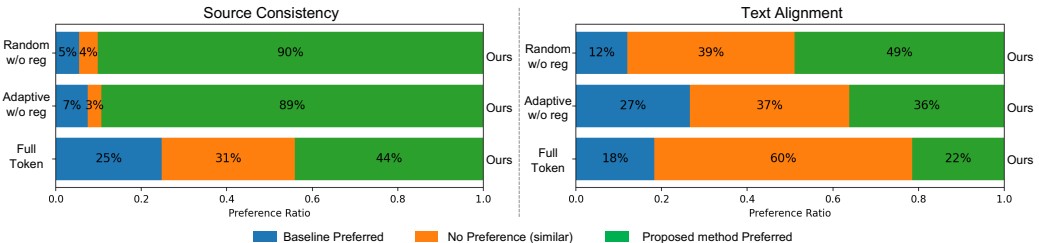

Figure 5: Human preference study results. The proposed method was consistently preferred over all baselines in terms of source consistency and was comparably preferred in terms of text alignment.

Figure 4 shows qualitative comparisons on PIEBench and GEdit-Bench. Our method achieves edits similar to the vanilla model while better preserving non-edit regions, as seen in the first two rows. Random masking, despite yielding the highest CLIP similarity on PIEBench and the best semantic alignment score on GEdit-Bench, often produces inconsistent results with over-smoothing or identity mismatches. Also, note that the semantic score of the proposed method is far from the those of source image, which indicates that the our approach effectively edits images. Categorical quantitative result and more qualitative examples are provided in the Appendix D.4 and D.5.

We conducted a human preference study to further assess the effectiveness of the proposed method for image editing (see protocol and details in Appendix C). The results in Figure 5 demonstrate that our approach substantially outperforms both random masking and adaptive masking without source regularization in terms of source consistency, with clear statistical significance, consistent with the qualitative comparisons. Importantly, for editing instruction alignment, our method achieves comparable or superior performance relative to each baseline, indicating that it preserves editing capability while improving efficiency. Overall, the proposed method delivers editing quality on par with the full-token vanilla model and, in many cases, provides better consistency in preserving non-edit regions and identity, with reduced computational cost.

## 4.2 IMPORTANCE OF SOURCE-CONSISTENCY REGULARIZATION

We next examine the role of source-consistency regularization by removing it from the flow ODE in Figure 6. When adaptive masking is applied without regularization ($\eta_t = 0$), source tokens are dropped without explicit compensation in subsequent flow ODE steps. This results in source information loss and degraded editing performance, manifested as reduced consistency and missing details in both quantitative and qualitative evaluations. Conversely, when source-consistency regularization is applied with random masking, the model tends to reproduce the source image instead of performing the intended edit, as explained by Proposition 1. Taken together, these results demonstrate that neither adaptive masking nor source-consistency regularization alone is sufficient; both components are necessary to achieve high-fidelity and instruction-consistent editing.

## 4.3 BEHAVIOR OF ADAPTIVE MASKING

Proposition 2 suggests that adaptive masks be determined by the residual between the clean estimate and the source image. In practice, however, the threshold must adapt to size and type of the edited region. On PIEBench, which provides ground-truth masks of non-edit regions, we observe a negative

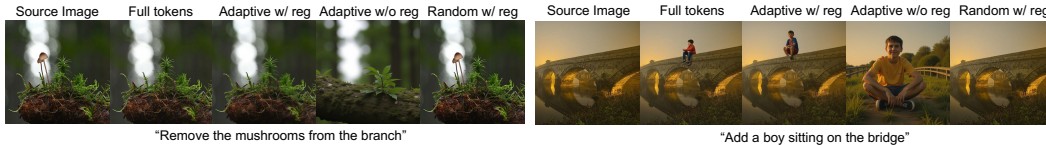

Figure 6: Ablation study on source consistency regularization. Only when adaptive token drop is applied with source consistency regularization, we obtain editing results similar to the vanilla model.

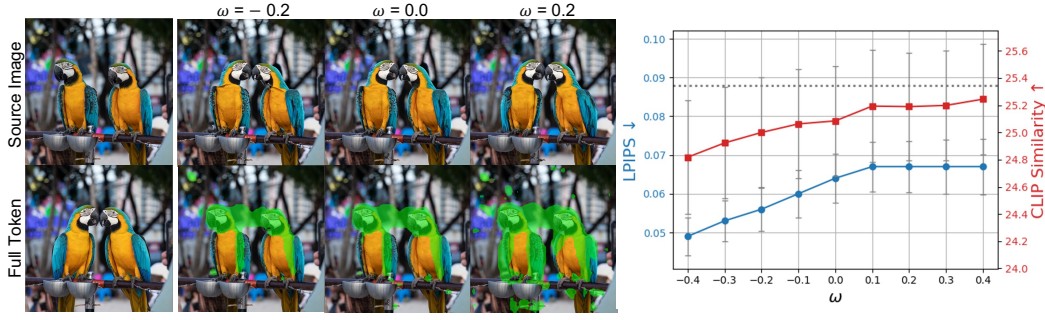

Figure 7: Effect of $\omega$ on the efficiency-performance trade-off. Larger $\omega$ retains more tokens, yielding performance closer to vanilla model but lowering efficiency. Left: qualitative examples with preserved tokens in green. Right: LPIPS ($\downarrow$) and CLIP similarity ($\uparrow$) versus $\omega$; dotted line denotes the full-token baseline. Reported are averages with error bars at $0.1\times$ standard deviation.

correlation between the drop ratio and edited area size (see Appendix D.1), indicating that triangle thresholding discards more tokens when edits are small and retains more when edits are large.

We further introduce a hyper-parameter $\omega$ to control the efficiency–performance trade-off by adjusting the threshold. Figure 7 illustrates its effect: the left subfigure highlights preserved tokens (in green), while the right plot reports LPIPS and CLIP similarity as functions of $\omega$. Increasing $\omega$ preserves more source tokens, yielding CLIP similarity close to the vanilla model but at reduced efficiency. In the shown example, preserved regions often align with edited areas such as the head or body of birds.

With small $\omega$, the mask concentrates on high-residual regions that are most critical for edits, while larger $\omega$ expands coverage to secondary areas, improving alignment with editing instructions. The right plot further indicates that smaller $\omega$ favors source consistency, whereas larger $\omega$ improves CLIP similarity at the cost of background LPIPS. Based on this trade-off, we set $\omega = 0.4$ for FluxKontext and $\omega = 0.1, 0.3$ for Qwen-Image-Edit. Notably, at matched CLIP similarity, our method attains higher source consistency than full-token editing, consistent with the additional qualitative results in Appendix D.5.

## 5 CONCLUSION

We propose a regularized flow ODE that replaces the computational burden of source image tokens, enabling partial token omission and yielding efficiency gains that scale with image resolution. The explicit regularization further ensures more consistent edits, supported by both quantitative and qualitative improvements. Our difference-based adaptive masking with triangle thresholding and standard deviation–based bias provides controllable trade-offs between efficiency and performance. As the method applies broadly to transformer-based flow models with conditioning tokens, it can be extended to higher-dimensional tasks such as image-to-video generation.

**Limitation** A limitation of the proposed approach is that it may discard tokens when the entire image is edited (e.g., in style transfer), where the vanilla model or random masking can be more effective. Nevertheless, the method is particularly well suited for localized edits and provides a principled framework for adaptive token pruning. More discussions are provided in Appendix D.4.

**Ethics Statement**    *TokenDrop* is proposed for efficient image editing, reducing computational cost while preserving editing capability and improving consistency in non-edited regions. As a result, the method may inherit potential negative impacts from the underlying editing model. There is a possibility that editing outcomes may deviate from the safe editing considerations learned during training, even though our approach modifies the sampling trajectory to enforce consistency with the source image.

**Reproducibility statement**    For reproducibility, we provide the complete proofs of the propositions in Appendix A, detailed information on the editing models, benchmarks, and evaluation metrics in Appendix B, and the full algorithms in Appendix 1 and Appendix 2. The code will also be released publicly.

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

# A   PROOF OF PROPOSITION 1 AND 2

**Proposition 1** (Pathwise Convergence with Random Mask). *Let $\boldsymbol{v}(\boldsymbol{z}) : \mathbb{R}^{L \times d} \to \mathbb{R}^{L \times d}$ be locally Lipschitz. With random binary mask $M_t \in \{0,1\}^{L \times d}$, a fixed source image latent $\boldsymbol{z}_{\mathrm{src}} \in \mathbb{R}^{L \times d}$ and time $t \in (0,1]$, the solution of expected-mask ODE in Eq. (12) $\boldsymbol{z}_t$ satisfies*

$$\lim_{t \to 0} \boldsymbol{z}_t = \boldsymbol{z}_{\mathrm{src}} \tag{13}$$

*Proof.* Let $m_t \in \{0,1\}^L$ be an i.i.d random mask with

$$\mathbb{P}[(m_t)_i = 1] = 1 - \lambda, \quad \lambda_t \in (0,1] \tag{16}$$

which is independent across $t$ and coordinates.

With the mask $M_t := m_t \mathbf{1}_d^\top \in \{0,1\}^{L \times d}$,

$$\mathbb{E}[M_t | \boldsymbol{z}_t] = \mathbb{E}[M_t] = (1 - \lambda)\mathbf{1}_{L \times d} \tag{17}$$

and

$$\mathbb{E}_{M_t}[\tilde{\boldsymbol{z}}_{0|t} | \boldsymbol{z}_t] = (1 - \lambda)\hat{\boldsymbol{z}}_{0|t} + \lambda \boldsymbol{z}^* \tag{18}$$

$$= (1 - \lambda')\hat{\boldsymbol{z}}_{0|t} + \lambda' \boldsymbol{z}_{\mathrm{src}} \tag{19}$$

where we use Eq. (10) for the last equality and $\lambda' = \frac{\eta_t \lambda}{1 + \eta_t} \in (0, \lambda]$ for $\eta_t > 0$.

Replacing the random mask with its expectation in Eq. (12) gives the ODE

$$\frac{d\boldsymbol{z}_t}{dt} = \frac{\boldsymbol{z}_t - \left((1 - \lambda')\boldsymbol{z}_{0|t} + \lambda' \boldsymbol{z}_{src}\right)}{t} = (1 - \lambda')\boldsymbol{v}(\boldsymbol{z}_t) + \frac{\lambda'}{t}(\boldsymbol{z}_t - \boldsymbol{z}_{src}) \tag{20}$$

By setting $\boldsymbol{k}_t = \boldsymbol{z}_t - \boldsymbol{z}_{src}$, we get

$$\frac{d\boldsymbol{k}_t}{dt} = (1 - \lambda')\boldsymbol{v}(\boldsymbol{z}_t) + \frac{\lambda'}{t}\boldsymbol{k}_t \tag{21}$$

Multiplying the integrating factor $\mu(t) = t^{-\lambda'}$ and computing integral from 1 to $t \in (0,1]$ yields

$$\boldsymbol{k}_t = t^{\lambda'}\boldsymbol{k}_1 + (1 - \lambda')t^{\lambda'}\int_1^t \tau^{-\lambda'}\boldsymbol{v}(\boldsymbol{z}_\tau)d\tau \tag{22}$$

where $\boldsymbol{k}_1 = \boldsymbol{z}_1 - \boldsymbol{z}_{\mathrm{src}}$ and $\boldsymbol{z}_1 \sim \mathcal{N}(0, \mathbf{I})$.

Suppose a radius $R > \|\boldsymbol{k}_1\| + 1$, and let

$$B_R := \{\boldsymbol{z} : \|\boldsymbol{z} - \boldsymbol{z}_{\mathrm{src}}\| \leq R\}, \quad \tau_R := \inf\{t \in (0,1] : \|\boldsymbol{k}_t\| = R\}. \tag{23}$$

For $t < \tau_R$, the trajectory stays inside $B_R$, so $\|\boldsymbol{v}(\boldsymbol{z}_\tau)\| \leq M_R = \sup_{\boldsymbol{z} \in B_R} \|\boldsymbol{v}(\boldsymbol{z})\| < \infty$. From the integral representation we then obtain

$$\|\boldsymbol{k}_t\| \leq t^{\lambda'}\|\boldsymbol{k}_1\| + (1 - \lambda')t^{\lambda'}\int_1^t \tau^{-\lambda'}\|\boldsymbol{v}(\boldsymbol{z}_\tau)\|d\tau \tag{24}$$

$$\leq t^{\lambda'}\|\boldsymbol{k}_1\| + (1 - \lambda')t^{\lambda'}M_R\int_1^t \tau^{-\lambda'}d\tau \tag{25}$$

where the first inequality holds as triangle inequality and the second inequality holds due to local Lipschitzness of $\boldsymbol{v}$, which implies boundedness on $B_R$.

Therefore,

$$\|\boldsymbol{k}_t\| \leq t^{\lambda'}\|\boldsymbol{k}_1\| + C(t, \lambda') \tag{26}$$

where

$$C(t, \lambda') = \begin{cases} t^{\lambda'}\|\boldsymbol{k}_1\| + M_R t^{\lambda'}\frac{1 - t^{1 - \lambda'}}{1 - \lambda'}, & \lambda' \in (0,1) \\ t\|\boldsymbol{k}_1\| + M_R t \log(1/t), & \lambda' = 1. \end{cases} \tag{27}$$

From Eq. (26), since Eq. (27) $\to 0$ as $t \to 0$, we conclude that

$$\lim_{t \to 0} \boldsymbol{z}_t = \boldsymbol{z}_{\mathrm{src}}. \tag{28}$$

$\square$

For the proof of the Proposition 2, we state preliminary results.

**Lemma 1** (Generalized Gronwall's Inequality (Teschl, 2012))**.** *Suppose $\psi(t)$ satisfies*

$$\psi(t) \le \alpha(t) + \int_0^T \beta(s)\psi(s)ds, \quad \forall t \in [0, T], \tag{29}$$

*with $\alpha(t) \in \mathbb{R}$ and $\beta(t) \ge 0$. Then,*

$$\psi(t) \le \alpha(t) + \int_0^t \alpha(s)\beta(s) \exp\left(\int_s^t \beta(r)dr\right) ds, \quad \forall t \in [0, T] \tag{30}$$

**Corollary 1.1.** *Suppose $\psi(t)$ satisfies*

$$\frac{\psi(t)}{dt} \le \gamma\psi(t) + \delta(t), \quad \forall t \in [0, 1] \tag{31}$$

*where $\gamma \ge 0$ denotes a constant, $\delta(t) \ge 0$ is integrable. Then, for $0 < \tau < t \le 1$,*

$$\psi(t) \le \exp(\gamma(t - \tau))\psi(\tau) + \int_\tau^t \exp(\gamma(t - s))\delta(s)ds \tag{32}$$

*Proof.* Integrate both sides from $\tau$ to $t$ yields

$$\psi(t) - \psi(\tau) \le \gamma \int_\tau^t \psi(r)dr + \int_\tau^t \delta(r)dr. \tag{33}$$

By plugging the following definitions to Lemma 1,

$$\alpha(t) := \psi(\tau) + \int_\tau^t \delta(r)dr \quad \text{and} \quad \beta(t) = \gamma \ge 0, \tag{34}$$

we get

$$\psi(t) \le \alpha(t) + \gamma \int_\tau^t \alpha(s)\exp(\gamma(t - s))ds. \tag{35}$$

Using integration by parts,

$$\int_\tau^t \alpha(s)\exp(\gamma(t - s))ds = [-\alpha(s)\exp(\gamma(t - s))]_\tau^t + \int_\tau^t \alpha'(s)\exp(\gamma(t - s))ds \tag{36}$$

$$= -\alpha(t) + \psi(\tau)\exp(\gamma(t - \tau)) \int_\tau^t \delta(s)\exp(\gamma(t - s))ds \tag{37}$$

where $\alpha(\tau) = \psi(\tau)$ and $\alpha'(s) = d\alpha(s)/ds = \delta(s)$ due to the first fundamental theorem of calculus. Therefore,

$$\psi(t) \le \psi(\tau)\exp(\gamma(t - \tau)) + \int_\tau^t \delta(s)\exp(\gamma(t - s))ds \tag{38}$$

$\square$

**Time convention.** We solve the flow ODE backward in time from 1 to 0. Accordingly, we use $t_1$ for the *initial* time and $t_0 < t_1$ for a *later* (smaller) time. Applying Corollary 1.1 with $(\tau, t) = (t_0, t_1)$ yields the reverse-time bound

$$\psi(t_0) \le e^{\gamma(t_1 - t_0)}\psi(t_1) + \int_{t_0}^{t_1} e^{\gamma(s - t_0)}\delta(s)\, ds, \tag{39}$$

Now, we can prove the Proposition 2 using Corollary 1.1.

**Proposition 2** (Error bound between two trajectories). *Let $\boldsymbol{v}(\boldsymbol{z}) : \mathbb{R}^{L \times d} \to \mathbb{R}^{L \times d}$ be locally Lipschitz. For any $t \in [t_0, t_1]$ with $0 < t_0 < t_1 \leq 1$, the error between trajectory $\boldsymbol{z}_{t_0}$ (Eq. (12)) and $\boldsymbol{z}_{t_0}^{\text{ref}}$ (Eq. (2)) satisfies*

$$\|\boldsymbol{e}_{t_0}\|^2 \leq C(t_1, t_0)\|\boldsymbol{e}_{t_1}\|^2 + \int_{t_0}^{t_1} \frac{C(t, t_0)}{t^2}\|\boldsymbol{r}_t\|^2 dt. \tag{14}$$

*where $\boldsymbol{e}_{t_0} = \boldsymbol{z}_{t_0} - \boldsymbol{z}_{t_0}^{\text{ref}}$, $C(t, t_0)$ is a time-dependent variable and $\boldsymbol{r}_t = \frac{\eta_t(1-M_t)\odot(\hat{\boldsymbol{z}}_{0|t}-\boldsymbol{z}_{\text{src}})}{1+\eta_t}$.*

*Proof.* $\qquad\qquad\qquad\qquad\qquad\qquad\qquad\qquad\qquad\qquad\qquad\qquad\qquad\qquad\qquad$ $\square$

Let the original flow ODE be

$$d\boldsymbol{z}_t^{\text{ref}} = \boldsymbol{v}(\boldsymbol{z}_t^{\text{ref}})dt. \tag{40}$$

Our regularized flow ODE is defined as

$$\frac{d\boldsymbol{z}_t}{dt} = \frac{\boldsymbol{z}_t - \tilde{\boldsymbol{z}}_{0|t}}{t}dt = \boldsymbol{v}(\boldsymbol{z}_t) + \frac{\eta_t}{1+\eta_t} \frac{(1-M_t)\odot(\hat{\boldsymbol{z}}_{0|t} - \boldsymbol{z}_{\text{src}})}{t} \tag{41}$$

$$= \boldsymbol{v}(\boldsymbol{z}_t) + \frac{\boldsymbol{r}_t}{t}. \tag{42}$$

Let the error between two trajectory be $\boldsymbol{e}_{t_0} := \boldsymbol{z}_{t_0} - \boldsymbol{z}_{t_0}^{ref}$. Take derivatives by $t_0$ yields

$$\frac{d\boldsymbol{e}_{t_0}}{dt_0} = \frac{d\boldsymbol{z}_{t_0}}{dt_0} - \frac{d\boldsymbol{z}_{t_0}^{ref}}{dt_0} = \left[\boldsymbol{v}(\boldsymbol{z}_{t_0}) - \boldsymbol{v}(\boldsymbol{z}_{t_0}^{ref})\right] + \frac{\boldsymbol{r}_{t_0}}{t_0}. \tag{43}$$

Suppose a common ball for two trajectories

$$B_R(c) := \{\boldsymbol{x} \in \mathbb{R}^{L \times d} : \|\boldsymbol{x} - \boldsymbol{c}\| \leq R\} \tag{44}$$

such that $\boldsymbol{z}_t^{\text{ref}}, \boldsymbol{z}_t \in B_R(c)$ for all $t \in [t_0, t_1]$. Since $\boldsymbol{v}$ is locally Lipschitz, there exists a constant $K_R < \infty$ such that $\|\boldsymbol{v}(\boldsymbol{x}) - \boldsymbol{v}(\boldsymbol{y})\| \leq K_R\|\boldsymbol{x} - \boldsymbol{y}\|, \forall \boldsymbol{x}, \boldsymbol{y} \in B_R$.

Then,

$$\frac{d}{dt_0}\|\boldsymbol{e}_{t_0}\|^2 = 2\boldsymbol{e}_{t_0}^\top \frac{d\boldsymbol{e}_{t_0}}{dt_0} \tag{45}$$

$$= 2\|\boldsymbol{e}_{t_0}\|\|\boldsymbol{v}(\boldsymbol{z}_{t_0}) - \boldsymbol{v}(\boldsymbol{z}_{t_0}^{ref})\| + \frac{2}{t_0}\boldsymbol{e}_{t_0}^\top \boldsymbol{r}_{t_0} \tag{46}$$

$$\leq 2K_R\|\boldsymbol{e}_{t_0}\|^2 + \frac{2}{t_0}\boldsymbol{e}_{t_0}^\top \boldsymbol{r}_{t_0} \tag{47}$$

$$\leq (2K_R + 1)\|\boldsymbol{e}_{t_0}\|^2 + \frac{\|\boldsymbol{r}_{t_0}\|^2}{t_0^2} \tag{48}$$

where the first inequality holds due to local Lipschitzness of $\boldsymbol{v}(\boldsymbol{z})$ in $B_R$ and the second inequality holds because $2\|a\|\|b\| \leq \|a\|^2 + \|b\|^2$.

By applying Corollary 1.1 with backward time, we get

$$\|\boldsymbol{e}_{t_0}\|^2 \leq C(t_1, t_0)\|\boldsymbol{e}_{t_1}\|^2 + \int_{t_0}^{t_1} C(t, t_0)\frac{\|\boldsymbol{r}_t\|^2}{t^2}dt \tag{49}$$

where $C(t, t_0) = \exp((2K_R + 1)(t - t_0))$

# B  IMPLEMENTATION DETAILS

In this section, we provide detailed information on the editing models and benchmarks used, along with further details on the adaptive mask.

**Editing Models**   We evaluate the proposed method using two editing models that employ Transformers as their backbone and incorporate source image tokens together with text and noise tokens in the input sequence. For both models, computations are performed with bfloat16 precision.

1. **Flux Kontext** (Batifol et al., 2025): We use the pre-trained checkpoint available at `https://huggingface.co/black-forest-labs/FLUX.1-Kontext-dev`. The classifier-free guidance scale is set to 3.5 and passed to the Transformer. Since the model is trained with guidance distillation, it predicts velocity once per timestep. We use 28 NFEs in our experiments.

2. **Qwen-Image-Edit** (Wu et al., 2025): We use the pre-trained checkpoint provided at `https://huggingface.co/Qwen/Qwen-Image-Edit`. The classifier-free guidance scale is set to 4.0 with 50 NFEs. This model uses Qwen-VL as a text encoder, which also takes the source image as input. We do not drop source image tokens from the encoder input, as they are used to incorporate source context and obtain a stronger representation of the editing instruction, rather than to provide explicit guidance for non-edit regions.

**Benchmarks**   We evaluate the proposed method on two image editing benchmarks:

1. **PIEBench** (Ju et al., 2023): We use the dataset provided at `https://github.com/cure-lab/PnPInversion`, which contains 700 synthetic and real images with corresponding source and target descriptions, editing instructions, and masks indicating non-edit regions.

2. **GEdit-Bench-EN** (Liu et al., 2025): We use the dataset available at `https://huggingface.co/datasets/stepfun-ai/GEdit-Bench`, which contains 600 synthetic and real images paired with editing instructions. For evaluation, we employ the pretrained Qwen2.5-VL 72B model (Bai et al., 2025) with prompts provided by (Liu et al., 2025), using only the English instructions.

**Definition of Metrics**   We use pre-defined metrics provided by each benchmark as following.

1. (Background) PSNR: we compute PSNR for non-edit region indicated by mask data.

2. (Background) LPIPS: we compute LPIPS for non-edit region indicated by mask data.

3. Structural Distance (i.e. DINO): we compute distance between features of entire image extracted by DINO_vitb8 variant.

4. (Background) MSE: we compute pixel-level mean-squared-error for non-edit region indicated by mask data.

5. (Background) SSIM: we compute SSIM for non-edit region indicated by mask data.

6. CLIP: we compute clip similarity between editing result and target prompt given by benchmark using clip-vit-large-patch14 variant.

7. CLIP-edit: we compute clip similarity for non-edit region indicated by mask data.

8. Q_SC: Semantic Consistency score between 0 to 10 is evaluated by Qwen2.5-VL 72B model to assess the degree to which the edited image is aligned to the given editing instruction.

9. Q_PQ: Perceptual Quality score between 0 to 10 is evaluated by Qwen2.5-VL 72B model to assess the naturalness of the edited image and the presence of artifacts.

10. Overall: The overall score is computed as the square root of the product of Q_SC and Q_PQ.

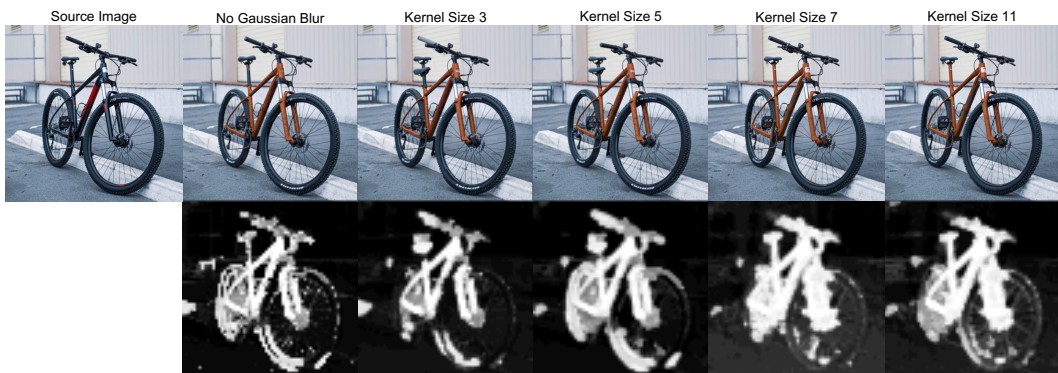

Figure 8: Effect of Gaussian blur to adaptive mask. Editing result with averaged mask along ODE timestep is displayed. Editing instruction: "Make the frame of the bike rusty".

**Adaptive Mask**  In Algorithm 2, we outline the process of adaptive mask generation. Empirically, applying a Gaussian blur to the difference map $D_t$ reduces noise and produces more stable masks that entirely cover edited region, as illustrated in Figure 8. We use a kernel size of 11 with a standard deviation of 1.0, followed by normalization of $D_t$ to the range $[0, 1]$. While these preprocessing steps may slightly alter the resulting mask, the fundamental principle of dropping source tokens with small deviations from the source image remains unchanged and plays a critical role in our method.

---

**Algorithm 2** AdaptiveMask

---

**Require:** Clean estimate $\hat{z}_{0|t}$, Source image latent $z_{\text{src}}$, ODE time $t$. Drop start time $t_d$
    **if** $t < t_d$ **then**
        $D_t \leftarrow |\hat{z}_{0|t} - z_{\text{src}}|$
        $D_t \leftarrow GaussianBlur(D_t)$
        $D_t = (D_t - \min(D_t))/(\max(D_t) - \min(D_t))$
        $\sigma_{D_t}^2 \leftarrow \mathbb{E}[\|D_t\|^2] - \mathbb{E}[D_t]^2$
        $\tau_\omega \leftarrow \text{TriangleThreshold}(D_t) - \omega\sigma_{D_t}$
        $M_t \leftarrow \mathbf{1}(D_t > \tau_\omega)$
    **else if** $t > t_d$ **then**
        $M_t = \mathbf{1} \in \mathbb{R}^L$
    **end if**
    **Return** $M_t$

---

## C  HUMAN PREFERENCE TEST

**Protocol**  To evaluate human preference on editing results, we conducted a user study. As illustrated in Figure 9, each participant was asked to choose either A or B to indicate their preference with respect to two criteria: alignment with the editing instruction and consistency with the source image. When participants could not identify a meaningful difference, they were allowed to select the "similar / not sure" option. Each participant completed 12 rounds. In each round, we provided the source image, the editing instruction, and two edited results: one produced by the proposed method and the other by a baseline. The baseline was randomly selected from the vanilla model, random drop with the same drop ratio, or adaptive drop without source consistency. The positions of our method and the baseline were randomized for each round. The preference survey was implemented using Google Apps Script. We recruited 23 participants, resulting in 276 total responses.

**Further Analysis**  In Figure 10, we present the human preference results for each editing model, FluxKontext and Qwen-Image-Edit, to provide further discussion. In terms of consistency, the proposed method outperforms alternative masking strategies for both models. Compared with the vanilla model, it significantly improves consistency in the case of Qwen-Image-Edit, while achiev-

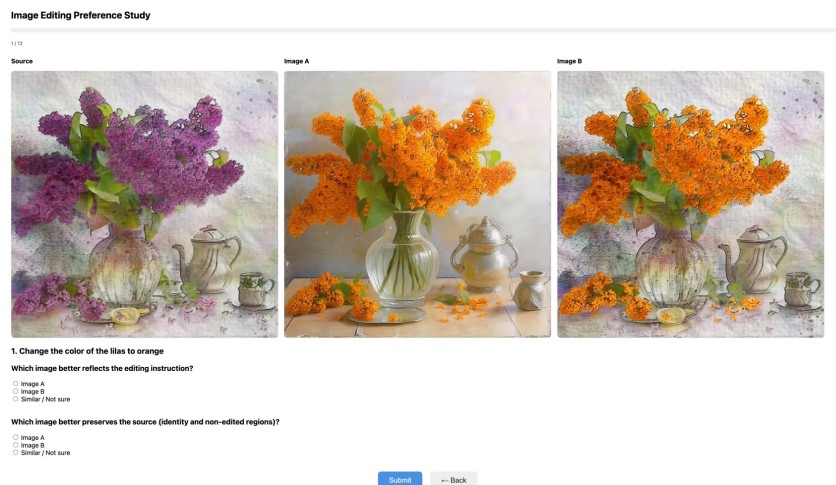

Figure 9: Human preference study interface.

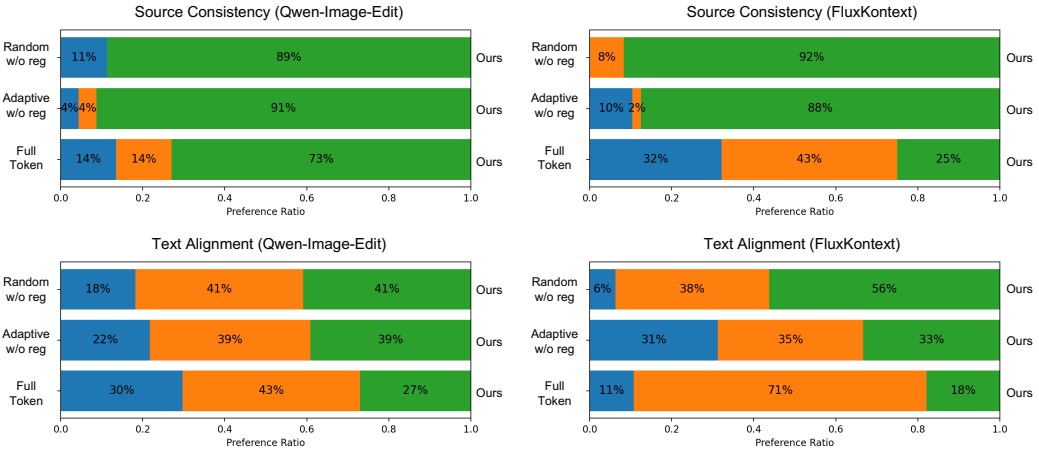

Figure 10: Human preference study result for each editing model.

ing comparable results with FluxKontext. This suggests that FluxKontext is inherently more robust in preserving consistency with the source image, whereas Qwen-Image-Edit leaves greater room for improvement. For text alignment, the results indicate that the proposed method preserves the behavior of the vanilla models while reducing computation cost. In both FluxKontext and Qwen-Image-Edit, the majority of responses fell into the "similar / not sure" category, showing that instruction-following remains largely unaffected. Where preferences were expressed, they were relatively balanced, with a slight edge toward our method in FluxKontext. These findings confirm that TokenDrop achieves efficiency gains without compromising the ability of the models to follow editing instructions.

# D ADDITIONAL RESULTS

## D.1 CORRELATION BETWEEN EDIT AREA AND MASK SIZE

We use triangle thresholding to adaptively determine the drop ratio from the difference between the clean estimate and the source image at each time step. Figure 11 shows a hexagonal heatmap of adaptive drop ratio versus editing mask size, where a larger mask corresponds to edits affecting larger regions. If the adaptive thresholding works correctly, it should assign smaller drop ratios when

the editing mask is large. The red trend line indeed shows a negative correlation, confirming that the proposed method effectively identifies appropriate thresholds.

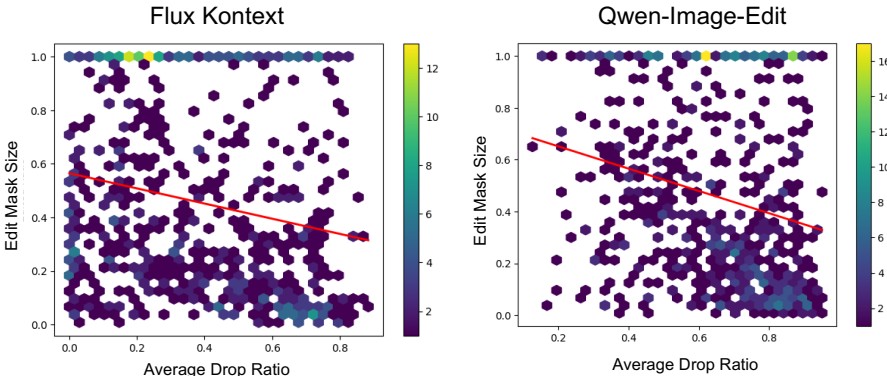

Figure 11: Hexagonal heatmap showing the correlation between editing mask size and average drop ratio. The red trend line indicates a negative correlation.

## D.2 HIGHER RESOLUTION IMAGE EDITING

To demonstrate the efficiency gains at higher resolutions, we evaluate image editing on $2048\times2048$ images from the DIV2K dataset (Ignatov et al., 2019) in Figure 12. Using the FluxKontext model, editing a 2K-resolution image requires 184 seconds. With the proposed method, the runtime is reduced to 91.69 seconds by dropping 89.5% of source tokens on average along sampling time. While accelerating the editing process, our method achieves results comparable to the vanilla model. Minor blurring is observed in the edited samples, which we attribute to the fact that FluxKontext was trained only at a fixed resolution of $1024\times1024$.

## D.3 EFFICIENT SAMPLING WITH LOW NFES

If the proposed method is combined with editing models that support fast sampling with only a few NFEs, efficiency can be further improved and editing time reduced to just a few seconds. To demonstrate this, we apply the method to FluxKontext with 8 NFEs, which is capable of robust image editing. We set $\omega = 0$ and use full-token sampling for the first two ODE steps. Table 2 reports the performance of the vanilla model and the proposed method on PIEBench. The proposed method improves consistency in non-edited regions, which is degraded when using a small number of NFEs, and reduces runtime by 21% with 65% drop ratio on average, while maintaining alignment with the target description. These results demonstrate that the method can also be applied to fast editing models that employ distillation to reduce NFEs.

| Model | Strategy | Runtime | PSNR ↑ | LPIPS ↓ | DINO ↓ | MSE ↓ | SSIM ↑ | CLIP ↑ | CLIP-edit ↑ |
|---|---|---|---|---|---|---|---|---|---|
| | Source Image | - | ∞ | 0.000 | 0.000 | 0.000 | 1.000 | 23.19 | 20.09 |
| Flux Kontext | Ours | **6.47** | **25.34** | 0.093 | **0.049** | **0.008** | **0.878** | 25.58 | 22.49 |
| | Full Tokens | 8.12 | 24.27 | **0.080** | 0.065 | 0.011 | 0.876 | **25.79** | **22.74** |

Table 2: Quantitative results of Flux Kontext on PIEbench. Even with 8 NFEs.

## D.4 PER-CATEGORY QUANTITATIVE RESULTS

To further analyze the performance of the proposed method, we report per-category quantitative results on GEdit-Bench in Figure 13. For most categories, our method (blue) achieves scores comparable to the vanilla model (orange) while significantly reducing runtime. However, in categories that require global changes, such as tone transformation or style transfer, performance decreases. As discussed in the limitation section, this stems from the thresholding mechanism, where certain tokens are consistently discarded.

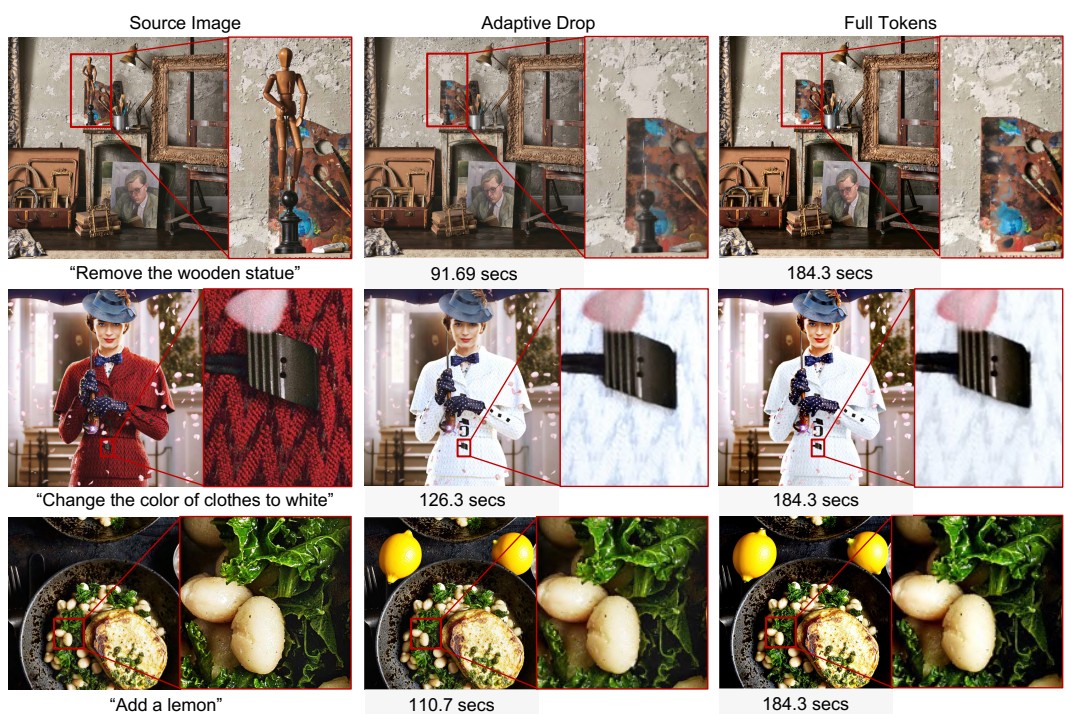

Figure 12: Image editing with 2k resolution using FluxKontext. The enlarged view corresponds to the red rectangle and highlights the consistency of the non-edit region with the source image.

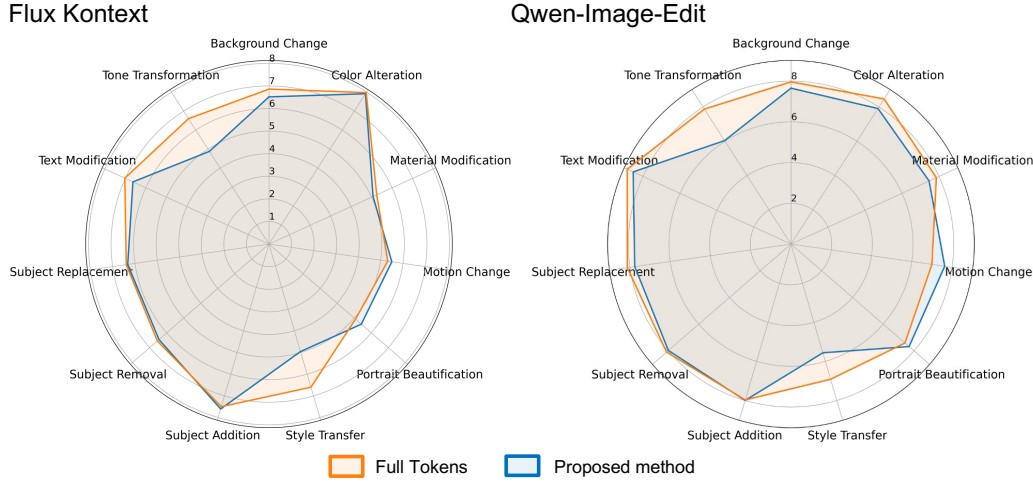

Figure 13: Quantitative result for each editing category of GEdit-Bench. Scores are evaluated by Qwen2.5-VL.

Localized editing, however, can still produce the intended results while reducing computational cost. For example, in the left column of Figure 14, our method edits only the woman and background partially, yet the output still reflects the editing instruction. Although these results are reasonable, the vanilla model achieves higher semantic scores such as CLIP similarity because it produces more extensive changes and aligns more closely with the instruction.

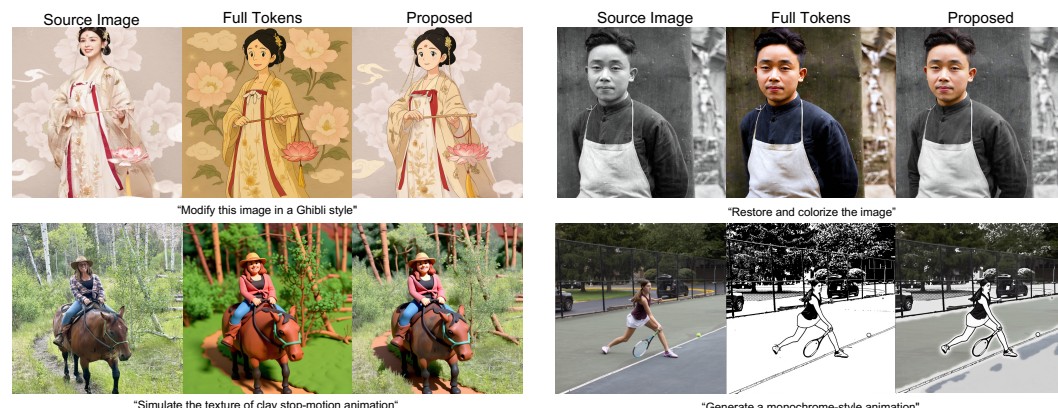

Figure 14: Localized editing with the proposed method. When a task requires full-image modification, threshold-based mask generation may lead to only partial edits. (Left) Localized editing achieves the intended result with improved efficiency. (Right) Localized editing fails to complete the edit.

There are also clear failure cases, as shown in the right column of Figure 14. For instance, the proposed method successfully colorizes the source image, but the effect remains confined to a limited region rather than the whole image.

For consistency in evaluation, we apply adaptive masking with regularization across all categories. In practice, however, the choice of dropping strategy can be adapted to the editing type. For example, random token dropping without source regularization performs well in this scenario, providing a simple and inexpensive alternative.

### D.5 ADDITIONAL QUALITATIVE RESULTS

In Figures 15-22, we present additional qualitative results obtained with FluxKontext and Qwen-Image-Edit on PIEBench and GEdit-Bench. Consistent with the main paper, the proposed method produces edits comparable to the vanilla model, and in some cases achieves higher fidelity in non-edited regions, while random token dropping with the same budget fails to preserve identity or consistency.

The right-most column shows the averaged adaptive masks over the flow ODE. Regions highlighted in blue denote preserved source tokens, while the remaining tokens are dropped and their information is compensated through the proposed regularized flow ODE. Notably, the adaptive mask concentrates on edited regions that exhibit large residuals relative to the source image. As the blue area decreases, computation becomes more efficient, yet the editing results remain nearly equivalent to those of the full-token baseline. Note that although the blue mask appears to cover almost every region, it represents an averaged mask and does not imply that most tokens are always preserved.

The editing instructions shown below each sample are taken directly from the dataset. Any typos originate from the dataset itself, and the editing model receives the instructions with these typos as input.

### E USAGE OF LLM

The LLM was used solely for grammar correction and improving the readability of the text. All conceptual contributions, analyses, and results were entirely developed by the authors.

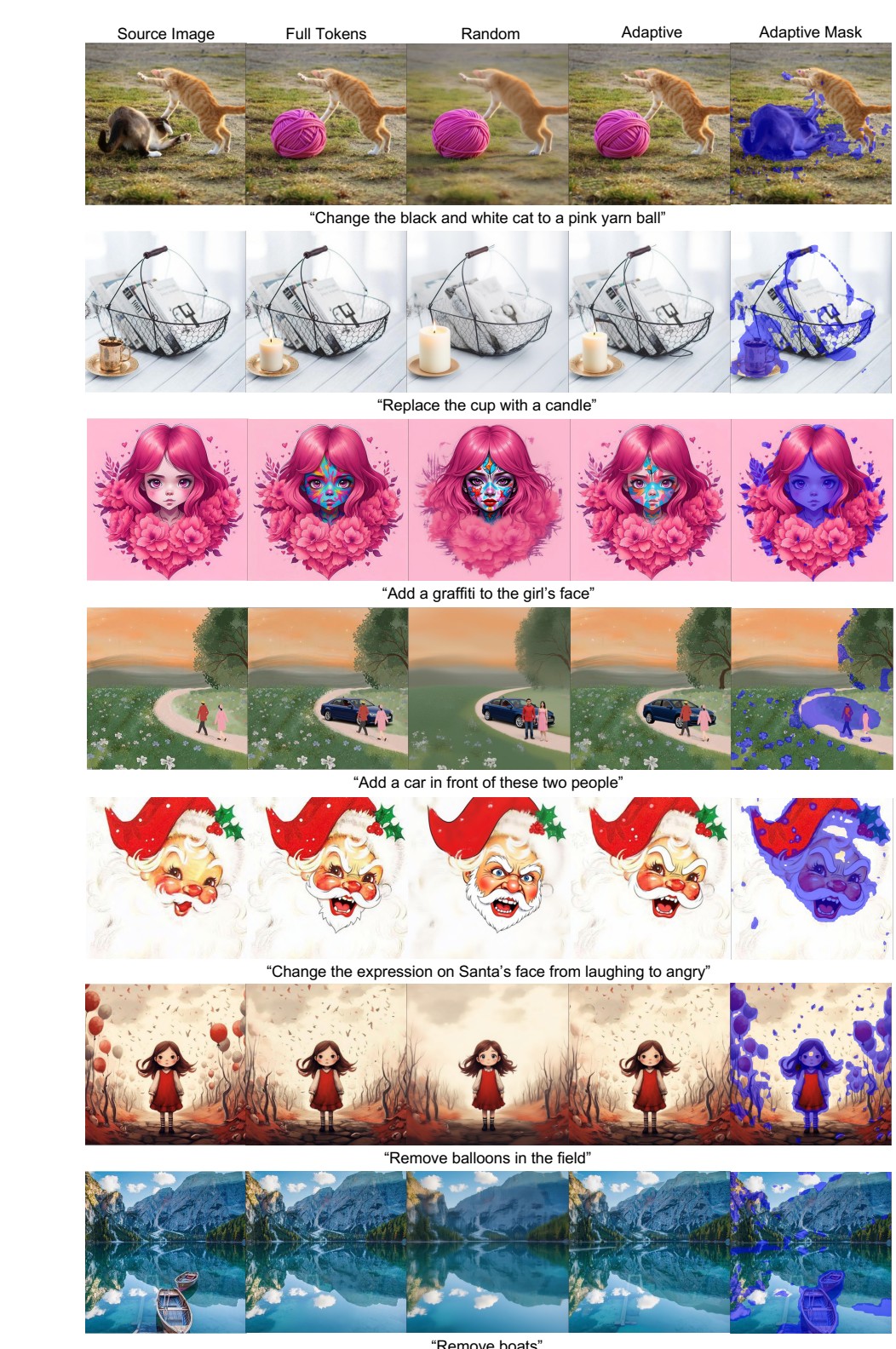

Figure 15: Qualitative results of FluxKontext on PIEBench. Image size: 1024×1024. The blue region denotes the averaged mask over sampling, while the actual masked region changes across steps.

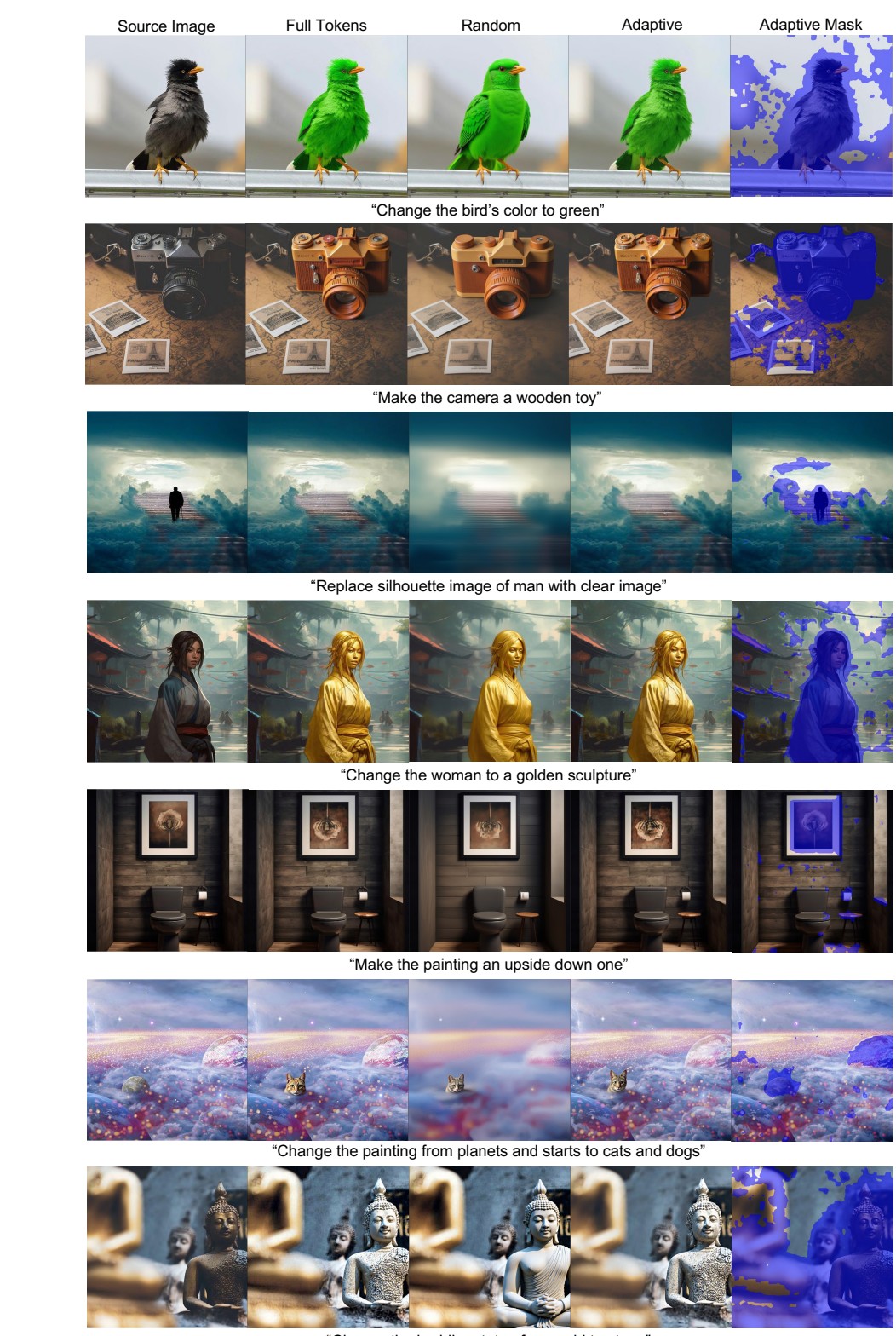

Figure 16: Qualitative results of FluxKontext on PIEBench. Image size: 1024×1024. The blue region denotes the averaged mask over sampling, while the actual masked region changes across steps.

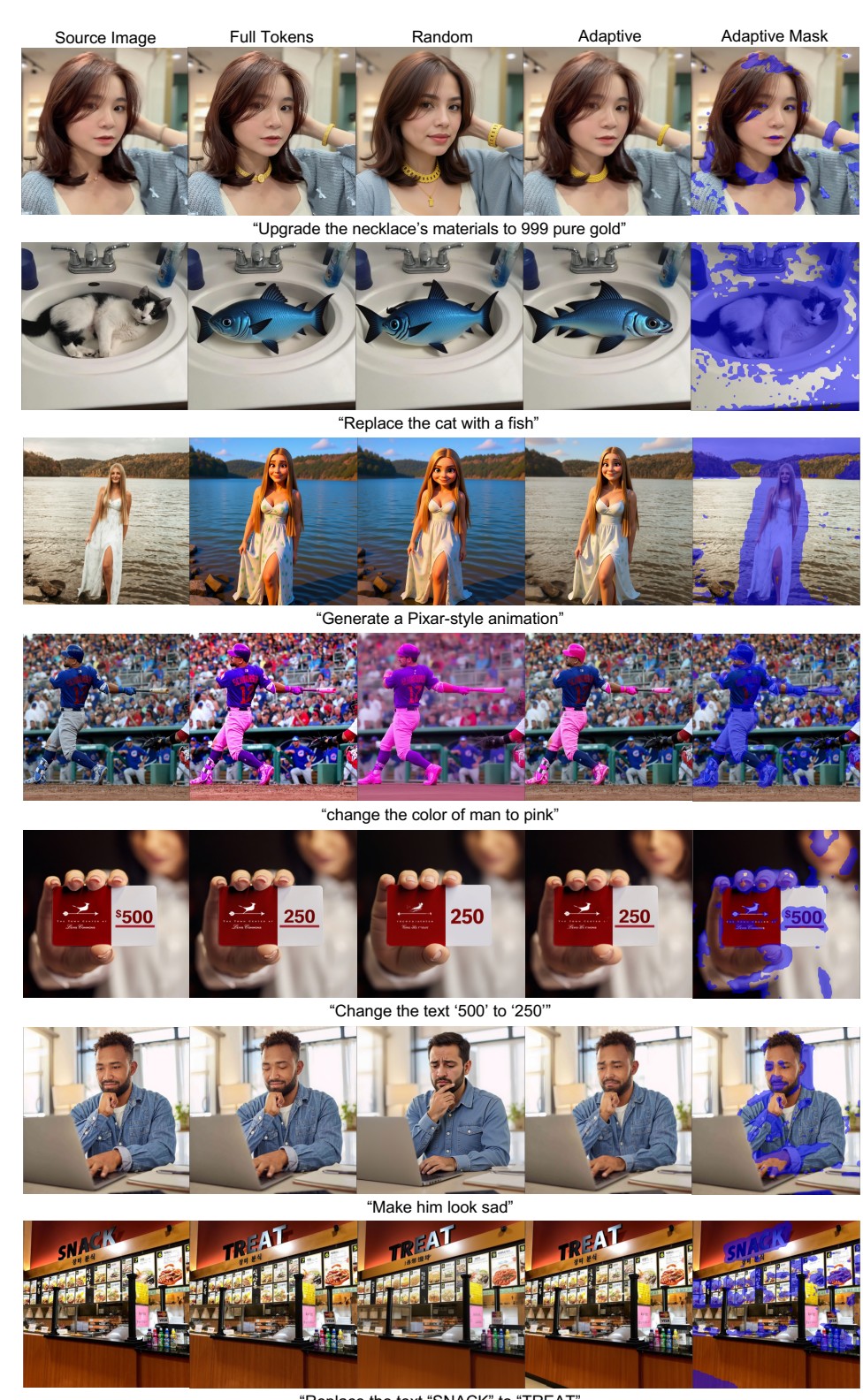

Figure 17: Qualitative results of FluxKontext on GEdit-Bench. Image size: 1024×1024. The blue region denotes the averaged mask over sampling, while the actual masked region changes across steps.

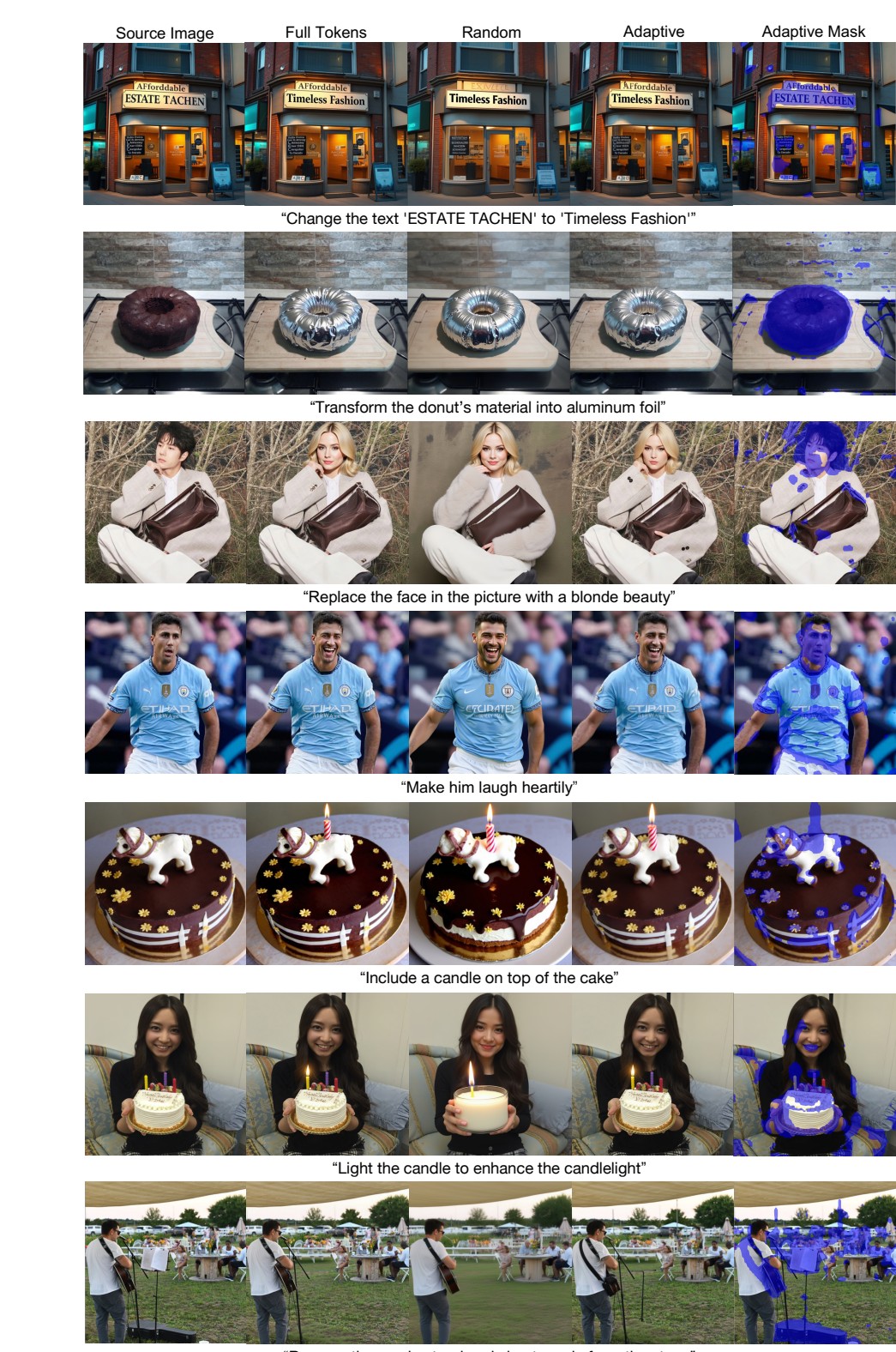

Figure 18: Qualitative results of FluxKontext on GEdit-Bench. Image size: 1024×1024. The blue region denotes the averaged mask over sampling, while the actual masked region changes across steps.

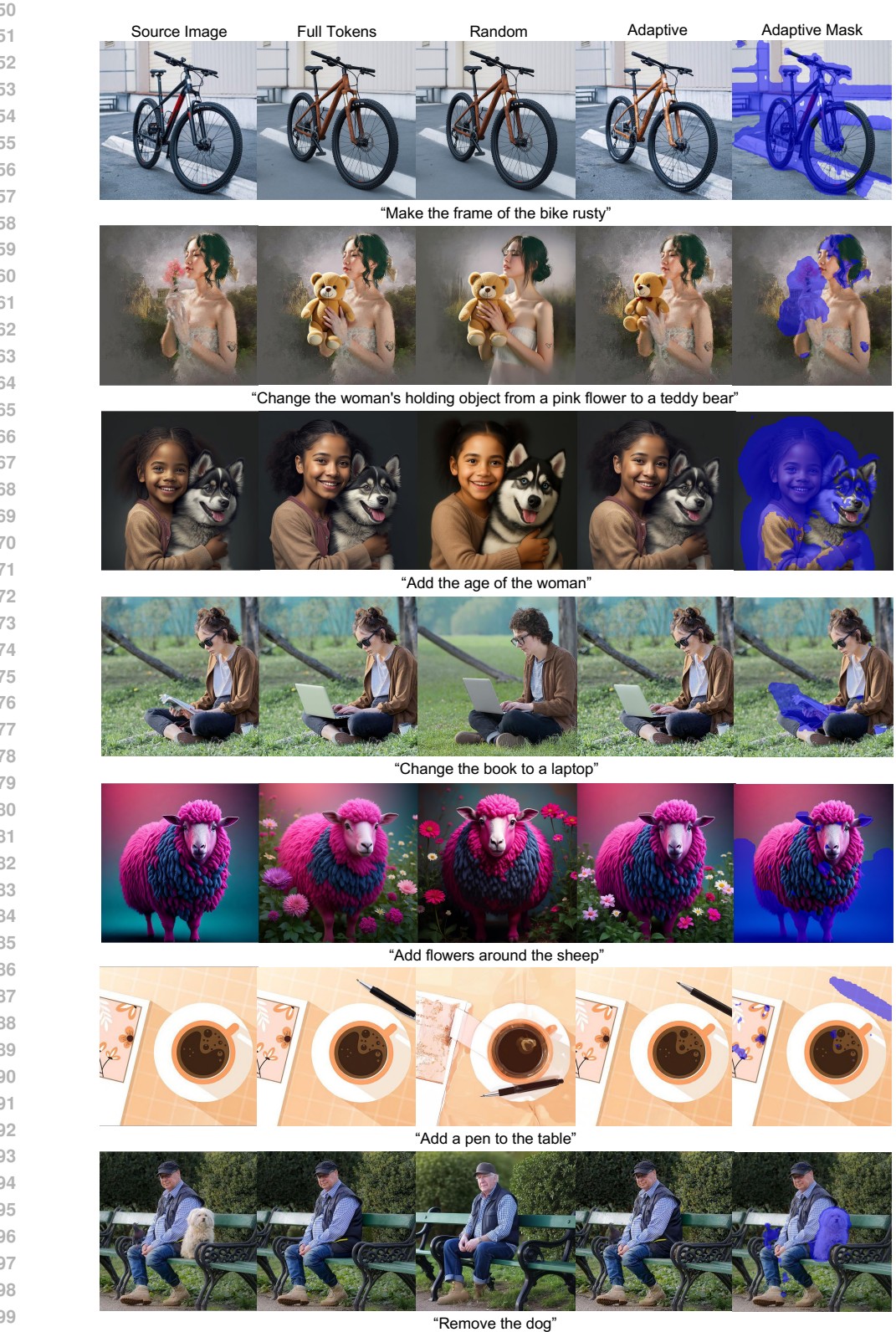

Figure 19: Qualitative results of Qwen-Image-Edit on PIEBench. Image size: 1024×1024. The blue region denotes the averaged mask over sampling, while the actual masked region changes across steps.

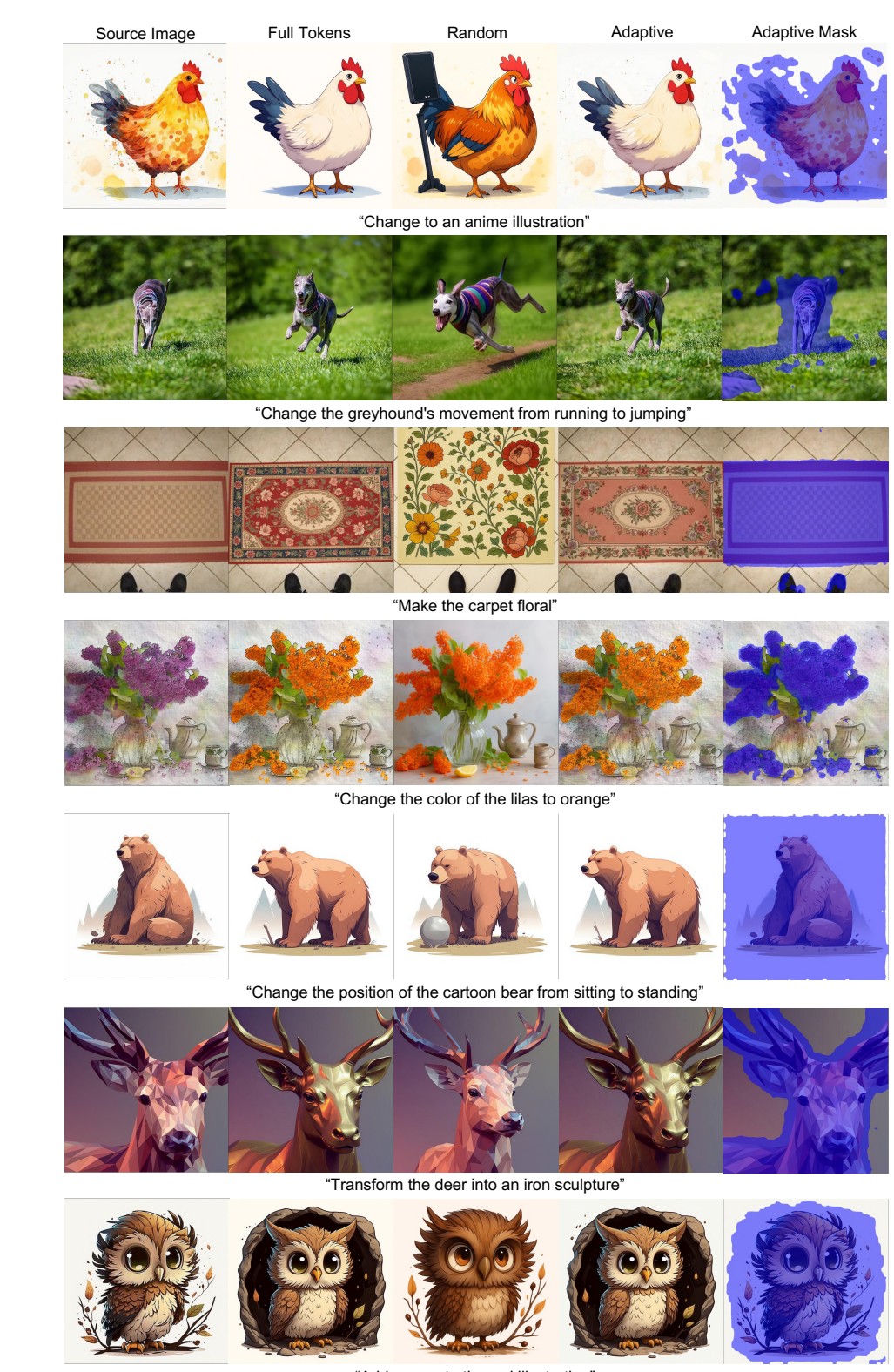

Figure 20: Qualitative results of Qwen-Image-Edit on PIEBench. Image size: 1024×1024. The blue region denotes the averaged mask over sampling, while the actual masked region changes across steps.

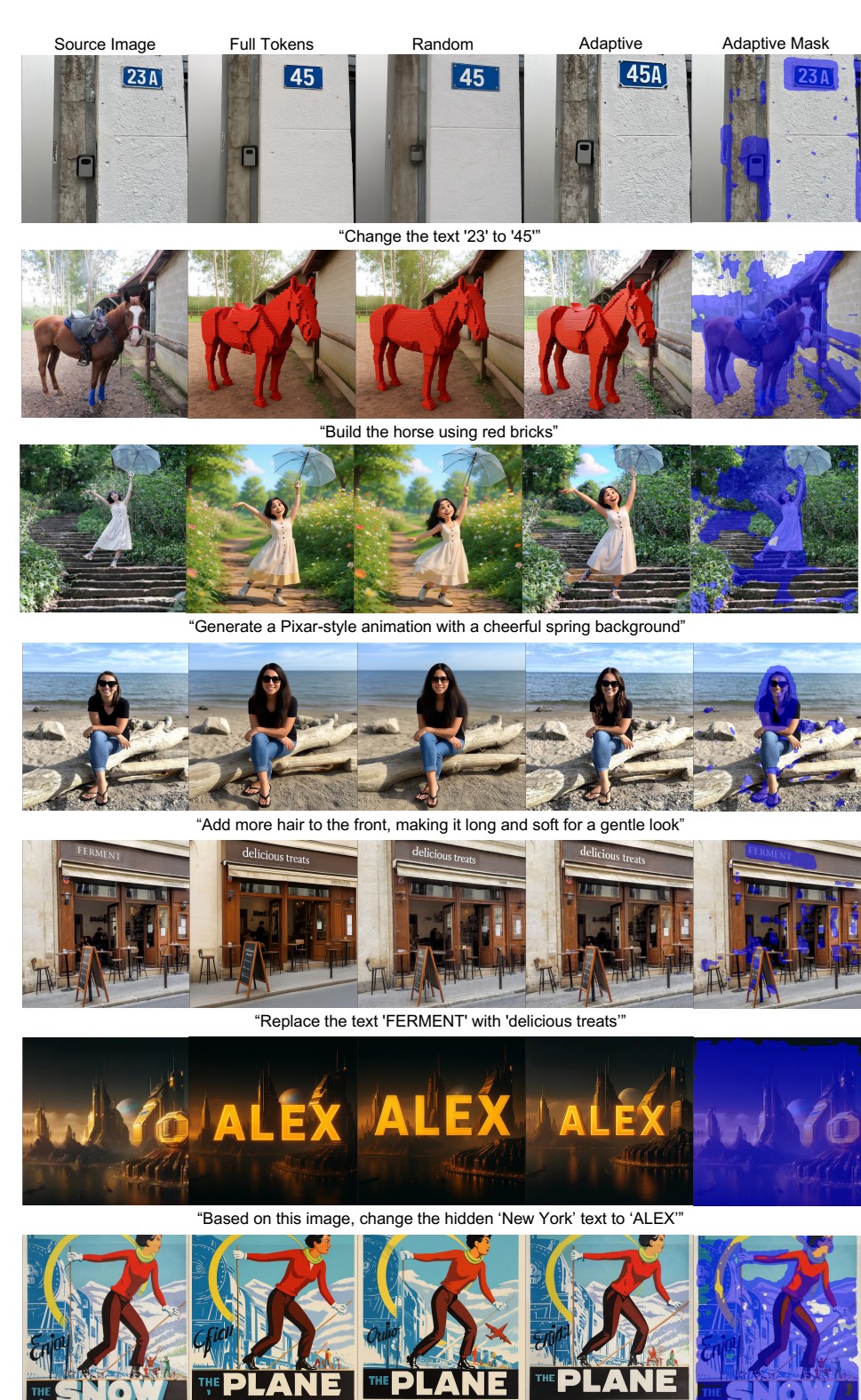

Figure 21: Qualitative results of Qwen-Image-Edit on GEdit-Bench. Image size: 1024×1024. The blue region denotes the averaged mask over sampling, while the actual masked region changes across steps.

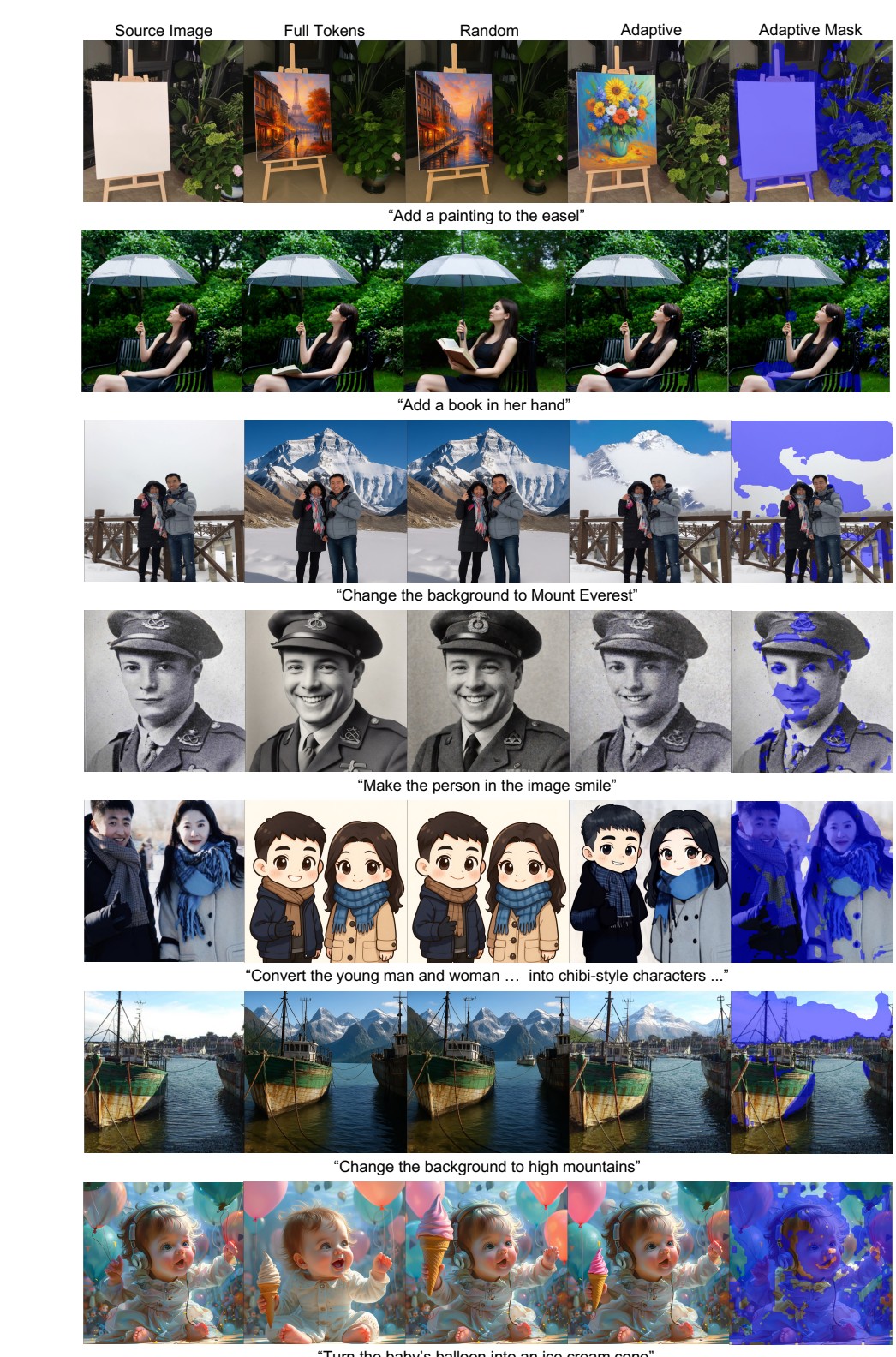

Figure 22: Qualitative results of Qwen-Image-Edit on GEdit-Bench. Image size: 1024×1024. The blue region denotes the averaged mask over sampling, while the actual masked region changes across steps.

