# OpenReview forum: "TokenDrop: Efficient Image Editing by Source Token Drop with Consistency Regularization"
_ICLR.cc/2026/Conference — ICLR 2026 Conference Withdrawn Submission_

### Official Review · Reviewer_nqGX · 2025-10-22

**Soundness:** 3
**Presentation:** 3
**Contribution:** 3
**Rating:** 4
**Confidence:** 4

**Summary:**

The paper presents TokenDrop, a dynamic token-pruning strategy for Diffusion Transformer (DiT)–based text-guided image editing models. The key idea rests on the assumption that image editing typically modifies only a small subset of visual tokens corresponding to localized semantic changes; thus, only the informative tokens should be preserved while redundant ones can be pruned. Technically, TokenDrop formulates the pruning process as a latent optimization problem, adding a time-varying regularization term to the flow ODE that enforces consistency with the source image at the locations of dropped tokens. This is adaptively guided by a mask computed from the difference between the source image and the predicted clean edit. Empirically, TokenDrop achieves 1.8–2.0× speed-ups over standard DiT models at high resolutions with negligible degradation in image-editing quality.

**Strengths:**

- The proposed idea is intuitive and well-motivated: viewing image editing as a sparse modification task naturally lends itself to token pruning, which is empirically validated through initial random-dropping experiments.
- The method is simple and efficient, admitting a closed-form update that significantly accelerates editing while maintaining overall editing quality.
- The authors demonstrate broad applicability by integrating TokenDrop into two strong DiT architectures, Flux Kontext and Qwen-Image-Edit, showing consistent runtime performance gains across setups.
- The paper includes ablation studies, analyzing the effects of the source-consistency regularization and the adaptive masking scheme, which strengthen the empirical case for the approach. A user study is also included to validate the proposed method.

**Weaknesses:**

– The related work discussion omits several recent efforts on efficient DiT architectures, especially token-sparsification and compression methods. For completeness, the paper should include and discuss pruning-based works such as SparseDiT (Chang et al., 2025), OminiControl (Tan et al., 2025), and DiffCR (You et al., 2025).
– The bias term ω in Eq. (15) appears to be empirically tuned per model or dataset, yet the paper does not offer a principled explanation or sensitivity analysis. A clearer theoretical or empirical justification for setting ω would improve reproducibility.
– In Proposition 1, the proof assumes that λ′ is constant when applying the integrating-factor step in Eqs. (21–22), whereas the text earlier defines λ′ as time-dependent. This mathematical inconsistency should be clarified.
– The approach’s limitations deserve more quantitative discussion, particularly its degradation on edits requiring global changes (e.g., style transfer), which could extend to cases where the manipulated region spans most of the image.

Missing References:
- Chang et al. SparseDiT: Token Sparsification for Efficient Diffusion Transformer. NeurIPS 2025.
- Tan et al. OminiControl2: Efficient Conditioning for Diffusion Transformers. arXiv preprint arXiv:2503.08280, March 2025.
- You et al. Layer- and Timestep-Adaptive Differentiable Token Compression Ratios for Efficient Diffusion Transformers. CVPR 2025.

**Questions:**

- Which model was used in the analysis of Figure 7, and how were the different ω values selected for each model or dataset? Did you observe large performance variance across configurations?
- The paper notes that performance deteriorates when the entire image must be edited (e.g., style transfer). Does this limitation also appear when the target object occupies a large portion of the frame, or only when full-frame edits are required?
- Could the authors comment on how TokenDrop interacts with attention-based acceleration or latent-space pruning methods. Are these approaches complementary or redundant?

---

### Official Review · Reviewer_p38N · 2025-10-27

**Soundness:** 3
**Presentation:** 2
**Contribution:** 1
**Rating:** 4
**Confidence:** 3

**Summary:**

The paper proposes TokenDrop, a training-free framework to accelerate text-guided image editing parameterized by a transformer-based diffusion models. The key idea is to utilize the difference between prediction and source latent to drop redundant source tokens during inference. To mitigate the information loss, the authors use a source consistency regularization term to the ODE formulation yielding a closed-form trajectory. Experiments demonstrate 22-100% speedup while maintaining editing quality on PIEBench and GEdit-Bench.

**Strengths:**

1. The paper tackles a practical problem to reduce the computation cost of multimodal transformer-based image editing.

2. The approach is lightweight and compatible with existing models without fine-tuning, making it appealing for practical usage.

3. The authors benchmark on multiple datasets with the state-of-the-art editing model to demonstrate the effectiveness. Several qualitative results are also presented.

4. The authors provide solid theorectical analysis on the flow ODE with regularized otpimization.

**Weaknesses:**

1. Although the authors claim that they use token dropping firstly for accelrating multi-modal image editing model, there has been exploration on reducing token number for image generation such as SiTo [1].

2. The insufficient evaluation is very limited to the proposed method and the base model. However, there are several applicable acceleration methods based on diffusion transformer such as ToCa [2] and L2C [3]. While it's not specifically designed for image editing model, it's strongly suggested that the authors should compare with these methods applicable to diffusion transformer.

[1] SiTo: Training-Free and Hardware-Friendly Acceleration for Diffusion Models via Similarity-based Token Pruning, AAAI 2025.
[2] Toca: Accelerating Diffusion Transformers with Token-wise Feature Caching, ICLR 2025.
[3] Learning-to-Cache: Accelerating Diffusion Transformer via Layer Caching, NeurIPS 2024.

**Questions:**

1. Could the authors provide illustration on which kinds of token are easier to be dropped?

2. Is there comparison to other methods to drop tokens apart from naive random drop?

---

### Official Review · Reviewer_omxK · 2025-11-01

**Soundness:** 3
**Presentation:** 2
**Contribution:** 3
**Rating:** 4
**Confidence:** 4

**Summary:**

This paper proposes TokenDrop, a training-free efficient image editing method designed to address the high computational overhead of transformer-based image editing models caused by excessive source image tokens. It achieves efficiency gains by adaptively dropping redundant source tokens based on the difference between clean estimates and source images, while introducing a source consistency regularization term into the flow ODE to compensate for information loss from token dropping. The method aims to balance inference speed and non-edited region consistency, and is validated on two mainstream transformer-based editing models across multiple benchmarks.

**Strengths:**

1. **Novel and Targeted Efficiency-Optimization Design**: TokenDrop directly targets the core bottleneck of transformer-based editing—redundant source tokens—and innovatively combines adaptive token selection with regularization-based information compensation. Unlike naive token dropping that sacrifices consistency, its adaptive masking (guided by clean estimate-source differences) and flow ODE reformulation effectively avoid non-edited region distortion, filling the gap between efficiency and consistency in existing methods.
2. **Strong Theoretical Foundation with Clear Mathematical Formalization**: The method rigorously reformulates the flow ODE as a latent optimization problem with a closed-form solution for the regularization term. Propositions on random mask convergence and trajectory error bounds provide solid theoretical support for adaptive masking and regularization, enhancing the credibility of the framework’s design logic.
3. **Comprehensive Ablation Studies Validate Component Efficacy**: Ablation experiments systematically verify the necessity of key components—source consistency regularization, adaptive masking, and Gaussian blur preprocessing for masks. By comparing variants without these components, the study clearly demonstrates how each module contributes to balancing speed and consistency, strengthening the persuasiveness of the method.
4. **Good Compatibility and Practicality**: As a training-free post-hoc optimization strategy, TokenDrop can be easily integrated into existing transformer-based flow models (e.g., Flux Kontext, Qwen-Image-Edit) without modifying model architectures or retraining. This compatibility, combined with its efficiency gains that scale with resolution, makes it highly applicable to real-world high-resolution editing scenarios.
5. **Rigorous Experimental Validation with Diverse Metrics**: Experiments cover multiple benchmarks (PIEBench, GEdit-Bench) and evaluation dimensions (quantitative metrics like PSNR/LPIPS, human preference studies). The inclusion of human evaluations (focusing on source consistency and text alignment) complements automatic metrics, providing a holistic assessment of editing quality that aligns with real-user needs.

**Weaknesses:**

1. **Insufficient Discussion on Global Editing Scenarios**: The method performs well for localized edits but struggles with global edits (e.g., style transfer, full-image tone transformation) due to its token-dropping logic that prioritizes non-edited regions. However, the paper lacks in-depth analysis of this limitation, such as why threshold-based masking fails for global tasks or potential mitigation strategies.
2. **Opaque Adaptive Mask Threshold Tuning**: While the paper introduces a bias term (ω) to control the efficiency-consistency tradeoff, it does not explain how to select ω for different editing tasks or models. The threshold-tuning process relies on manual adjustment, lacking an adaptive mechanism, which may hinder its usability in practical deployment.
3. **Incomplete Comparison with State-of-the-Art Efficiency Methods**: The paper only compares TokenDrop with naive random token dropping and full-token baselines, but omits comparisons with other advanced efficiency-optimized editing methods. This makes it difficult to fully assess its competitiveness in the broader efficiency-optimization landscape.

**Questions:**

Please refer to the detailed points I raised in the "Weakness" section and respond to each numbered item in your rebuttal with clarifications.

---

### Official Review · Reviewer_6efM · 2025-11-02

**Soundness:** 3
**Presentation:** 3
**Contribution:** 3
**Rating:** 6
**Confidence:** 2

**Summary:**

The paper presents TokenDrop, a training-free method to accelerate transformer-based image editing models. The proposed solution is to adaptively drop a portion of the source image tokens during the inference process. To mitigate the information loss and preserve quality in non-edited regions, the authors reformulate the standard flow ODE sampling as a regularized optimization problem. The method is shown to achieve significant speedups (up to 2x at 2048px resolution) across different models while maintaining high editing quality and better preserving non-edited regions.

**Strengths:**

- Novelty. The paper offers a novel solution to speed up the image editing model inference without performance degradation. The key novelty lies in the reformulated the flow ODE and the introduction of the source consistency regularization. The method is training free which is another advantage.

- Comprehensive evaluation: The effectiveness of TokenDrop is demonstrated thoroughly. The authors test their method on two different state-of-the-art models (Flux Kontext and Qwen-Image-Edit) and two different benchmarks (PIEBench and GEdit-Bench) with extensive quantitative metrics. A human study is also provided.

**Weaknesses:**

- Limited for global edits. The method's core strength is preserving non-edited regions by dropping their corresponding tokens. As the authors acknowledge, this makes it less suitable for edits that require global changes, such as style transfer, where most tokens should be edited and thus fewer can be dropped.

**Questions:**

L314: we use all tokens during the first few iterations to get a better initial mask estimation. How is the number of "initial iterations" determined, and how sensitive is the final result to this choice?

---

### Note · Authors · 2025-11-14

I have read and agree with the venue's withdrawal policy on behalf of myself and my co-authors.